# DNA strand breaks and gaps target retroviral intasome binding and integration

Gayan Senavirathne [1], James London[1], Anne Gardner [1], Richard Fishel [1,2] ✉ & Kristine E. Yoder [1,2,3] ✉

Retrovirus integration into a host genome is essential for productive infections. The integration strand transfer reaction is catalyzed by a nucleoprotein complex (Intasome) containing the viral integrase (IN) and the reverse transcribed (RT) copy DNA (cDNA). Previous studies suggested that DNA target-site recognition limits intasome integration. Using single molecule Förster resonance energy transfer (smFRET), we show prototype foamy virus (PFV) intasomes specifically bind to DNA strand breaks and gaps. These break and gap DNA discontinuities mimic oxidative base excision repair (BER) lesion-processing intermediates that have been shown to affect retrovirus integration in vivo. The increased DNA binding events targeted strand transfer to the break/gap site without inducing substantial intasome conformational changes. The major oxidative BER substrate 8-oxo-guanine as well as a G/T mismatch or +T nucleotide insertion that typically introduce a bend or localized flexibility into the DNA, did not increase intasome binding or targeted integration. These results identify DNA breaks or gaps as modulators of dynamic intasome-target DNA interactions that encourage site-directed integration.

Retroviruses cause clinically significant diseases such as AIDS or leukemia and must integrate their viral genomes into a host cellular genome to replicate[1]. Integration is catalyzed by a poorly understood retroviral pre-integration complex (PIC)[2,3]. In addition to cellular factors, the PIC contains viral integrase (IN) multimers in complex with the two long terminal repeat (LTR) sequences that flank the viral cDNA collectively known as an intasome[4]. As an intasome component, IN initially excises two nucleotides (nt) from the LTR[5] producing recessed 3'-OHs[4], and then catalyzes two consecutive $S_N2$ strand transfer reactions that covalently link these recessed ends across one major groove of a target DNA separated by 4–6 base pairs (bp), depending on the retrovirus genera[4,6,7]. The integrated cDNA is flanked by 4–6 nt gaps of host sequence and 2 nt 5'-flaps of viral DNA that are restored to fully duplex DNA by the host DNA repair machinery[8–11].

Alignment of numerous integration sites suggests that most retroviruses prefer to integrate at symmetrical but degenerate consensus DNA sequences that for HIV-1 often positions guanine (G) adjacent to the strand transfer sites (T D G * (G/V) T N A (C/B) * C H A; * strand transfer site)[12,13]. A large-scale siRNA screen identified base excision repair (BER) components as the most frequent DNA repair genes that altered HIV-1 integration efficiency[8]. Of these, deletion of oxidative damage BER glycosylases *OGG1* or *MUTYH* reduced HIV-1 integration as well as the preference for G and/or disfavored C adjacent to the strand transfer site[11]. The OGG1 glycosylase principally excises 8-oxo-guanine (8-oxo-G) residues in DNA that are the most frequent base damage in cells[14,15]. The MUTYH glycosylase excises adenine (A) residues that are frequently misincorporated across from an 8-oxo-G lesion during replication[16]. Deletion of the BER polymerase β (*Polβ*) also significantly decreased HIV-1 integration[17]. However, the *Polβ* phenotype was linked to the 5' deoxyribose phosphate (5' dRP) lyase activity and not its polymerase activity[17]. The 5' dRP lyase removes the deoxyribose sugar and associated phosphate leaving a 1 nt gap in the DNA after base excision by a glycosylase[18–20]. Interestingly, bifunctional

[1]Department of Cancer Biology and Genetics, The Ohio State University College of Medicine, Columbus, OH 43210, USA. [2]Molecular Carcinogenesis and Chemoprevention Program, The James Comprehensive Cancer Center and Ohio State University, Columbus, OH 43210, USA. [3]Center for Retrovirus Research, The Ohio State University, Columbus, OH 43210, USA. ✉e-mail: fishel.7@osu.edu; yoder.176@osu.edu

OGG1 also contains an intrinsic 5′ dRP lyase, while other monofunctional glycosylases generally utilize the abasic endonuclease APE1 to introduce the strand break required to complete BER[16].

Historical studies have suggested that bent or flexible DNA is a favored target for retroviral integration in vivo[21,22] and in vitro[23–30]. Prototype foamy virus (PFV) is a member of the *spumavirus* subfamily, and its IN has been extensively studied since it shares similar catalytic site geometry, chemistry, and therapeutic inhibitor sensitivity with the pathogenic lentivirus HIV-1 IN[31–33]. Structures of the PFV intasome target capture complex (TCC) and the strand transfer complex (STC) have been solved and represent key steps that provide a biophysical window into retroviral integration[23,31]. Moreover, real-time single molecule imaging has detailed the dynamic interactions between the PFV intasome and target DNA[24,28,30] and concluded that DNA site recognition limited PFV integration[24]. Here we have utilized single molecule Förster resonance energy transfer (smFRET)[34] to probe the real-time interactions between PFV intasomes with DNA containing a variety of DNA lesions and BER pathway intermediates that introduce localized bends or flexibility into DNA[35–38]. We found that PFV intasomes bind specifically to target DNA containing single stranded breaks or gaps resulting in site-specific integration; a unique observation for retroviruses.

## Results

### PFV intasomes bind to single strand DNA breaks and gaps within a target DNA

PFV intasomes were assembled with pre-processed recessed 3′-OH viral U5 oligonucleotides (vDNA) containing a Cy3 FRET-donor fluorophore on the non-transferred strand 11 bp from the 3′-OH of the transferred strand (Cy3-PFV; Supplementary Table 1). Concerted Cy3-PFV intasome integration into a supercoiled target DNA recapitulated unlabeled PFV intasomes, confirming that the Cy3-labeled vDNA does not affect integration activity (Supplementary Fig. 1)[24].

A 60 bp target DNA that was fully duplex or contained defined lesions was synthesized containing a Cy5 FRET-acceptor fluorophore located on the undamaged strand, 10 or 11 bp to the 5′-side relative to the lesion site, and an AlexaFluor488 (AF488) marker fluorophore on the damaged strand 3 bp from the 3′ end (Supplementary Table 1). This target DNA was bound to a passivated smFRET flow cell surface at the 5′-end of the lesion-containing strand, purified Cy3-PFV intasomes were infused and illuminated with a 532 nm laser (Fig. 1a). Binding by the Cy3-PFV intasomes near the lesion resulted in Cy5 (660 nm) FRET emission consistent with dynamic TCC interactions (Fig. 1b, c, left; 100 ms frame rate). Background correction and Hidden Markov Modeling (HMM) generated a transition density plot[39] that was used to determine FRET efficiency (E) and reversibility (Fig. 1c, middle; see Supplementary Fig. 2a for the analysis algorithm and Supplementary Fig. 3a for transition density plot). The total number of FRET events (n) from multiple target DNA molecules (N) were aligned to the PFV intasome infusion time to produce a post-synchronized histogram (N = 552, n = 46; Fig. 1c, right)[40]. A narrow range of Cy5 fluorescence (E_{pseudo} ~ 0.06) was detected in both the transition density plots and post-synchronized histograms. After inspection of the real-time movies these were determined to represent excursions of likely aggregated free Cy3-PFV intasomes across the evanescent field containing the target DNA (Supplementary Movie 1). Such excursions were easily recognized since they saturated the Cy3 channel and then bled into the Cy5 channel producing a pseudo-FRET signal (Fig. 1c, right). The frequency of these pseudo-FRET events did not change with DNA substrate or frame rate, in agreement with an intrinsic fluorescent background element (Supplementary Tables 2 and 3). Rare higher FRET events (E > 0.1) were detected that appeared to correspond to PFV intasome binding and dissociation events near the Cy5 fluorophore on the target DNA (Fig. 1c, right). A similar pattern was observed when an 8-oxo-G, a G/T mismatch or +T nucleotide insertion

was present in the target DNA (see: transition density plots in Supplementary Fig. 3a–d; and post-synchronized histograms in Supplementary Fig. 4a–c). The lack of a substantial number of FRET events above the pseudo-FRET background, suggests very little if any specific PFV intasome binding to these target DNA substrates.

In contrast, a target DNA substrate containing a 1 nt gap with a 3′-OH and 5′-phosphate [1 nt Gap (5′-P)] similar to a 5′-dRP lyase BER intermediate, consistently displayed bursts of Cy3-PFV binding events accompanied by Cy5 FRET events (Fig. 1d, left). Inspection of real-time movies confirmed that these FRET events represented PFV intasome-lesion interactions arising from anti-correlated colocalized diffraction-limited Cy3-Cy5 spots (Supplementary Movie 2). The FRET bursts from individual traces collected at 100 ms frame rate (Fig. 1d, center) converged into a distinct ensemble FRET peak that was clearly separate from the pseudo-FRET background ($E_{1nt\ Gap\ (5′-P)}$ ~ 0.16 ± 0.05; N = 540, n = 3027; Fig. 1d, right; Table 1). A comparison with calculated FRET (Fig. 1b) suggests these binding events reflect the formation of a TCC, where the PFV intasome forms a stable but transient binding complex near the DNA lesion (Fig. 1b, d). Target DNA substrates containing a single-strand scission [Nick (5′-P); $E_{Nick\ (5′-P)}$ ~ 0.15 ± 0.08; N = 549, n = 1084], a 2 nt gap [2 nt Gap (5′-P); $E_{2nt\ Gap(5′-P)}$ ~ 0.18 ± 0.10; N = 474, n = 1175] as well as a 5′-phosphate free nick ($E_{Nick\ (5′-OH)}$ ~ 0.17 ± 0.07; N = 558, n = 507), 1 nt Gap ($E_{1nt\ Gap\ (5′-OH)}$ ~ 0.19 ± 0.05; N = 496, n = 2784) and 2 nt Gap ($E_{2nt\ Gap\ (5′-OH)}$ ~ 0.23 ± 0.11; N = 540, n = 799) also yielded significant FRET events, which appear consistent with the formation of a TCC (Table 1; see: transition density plots in Supplementary Fig. 3h–g; and post synchronized histograms in Supplementary Fig. 4d–h). As might be expected, the number of FRET transitions per molecule increased with DNA substrates that displayed specific binding events (Supplementary Fig. 5). The uniform peak of FRET efficiency suggests the Cy3-PFV intasomes bind target DNA within a narrow window around the DNA lesions.

To quantify the relative binding efficiency of PFV intasomes to the DNA lesions, we normalized the frequency of TCC FRET binding events to the pseudo-FRET excursion events, which appear generally constant at a fixed Cy3-PFV intasome concentration and frame rate between the various target DNA substrates (Supplementary Tables 2 and 3). As anticipated, there were few FRET events outside the normalized pseudo-FRET peak when duplex DNA, DNA containing an 8-oxo-G/C lesion, a G/T mismatch or a +T nucleotide insertion were examined (E > 0.1; Fig. 1e; for individual gaussian fits see: Supplementary Fig. 6a–d). A similar normalized TCC FRET histogram pattern was observed utilizing 1 s frame rate data (see: Supplementary Fig. 7a–d for post-synchronized histograms; Supplementary Fig. 7k for normalized FRET histograms; and Supplementary Table 3 for pseudo-FRET statistics at 1 s frame rate). Because the G/T mismatch introduces significant local flexibility into the DNA[41,42] and the +T insertion produces a relatively stable 22° DNA bend[43], these observations suggest that the preference for flexible or bent DNA by retrovirus intasomes[20,43–45] may be more subtle than previously appreciated.

In contrast, a nick, 1 nt gap or 2 nt gap, with or without the 5′-P, displayed normalized TCC FRET peaks that were clearly separated from the pseudo-FRET background (Fig. 1f, g; see: Supplementary Fig. 6e–j for individual gaussian fits; and Supplementary Fig. 4d–h for post synchronized histograms). The relative frequency of these TCC events suggested that Cy3-PFV intasomes preferred a 1 nt gap » nick ≥ 2 nt gap. We noted a broadening of the normalized TCC FRET peak as well as an increased average FRET efficiency with target DNA containing a 2 nt gap (Fig. 1f, g). Because the 2 nt gap extends further toward the Cy5 FRET acceptor, these results suggest an increase in the number of asymmetric binding events that on average bind 5′ of the gap and therefore closer to the Cy5 fluorophore. A similar pattern for the normalized TCC histograms was observed utilizing the 1 s frame rate data, reducing the possibility of restriction bias (see: Supplementary Fig. 7e–j for post-synchronized histograms; Supplementary

Fig. 7l, m for normalized FRET histograms; and Supplementary Table 3 for pseudo-FRET statistics at 1 s frame rate).

We utilized HMM to determine the binding ($\tau_{on}$) and dissociation ($\tau_{off}$) kinetics of PFV intasomes (see: Supplementary Fig. 8a for analysis algorithm). As expected, the non-pseudo-FRET binding events to an 8-oxo-G/C, a G/T mismatch or a +T nucleotide insertion were extremely rare, often < 1 event per molecule over the 3 min observation window

(see Supplementary Fig. 4a–c for post-synchronized histograms 100 ms frame rate; Supplementary Fig. 5a–d for counts per molecule; Supplementary Fig. 7a–d for post-synchronized histograms 1 s frame rate), making any $\tau_{on}$ and $\tau_{off}$ values statistically insignificant (Supplementary Fig. 8b). However, the 1 nt gap (5'-P) and 1 nt gap (5'-OH) experienced an average of ~6 binding events per molecule with a distribution of 1–14 binding events for the vast majority of molecules (see

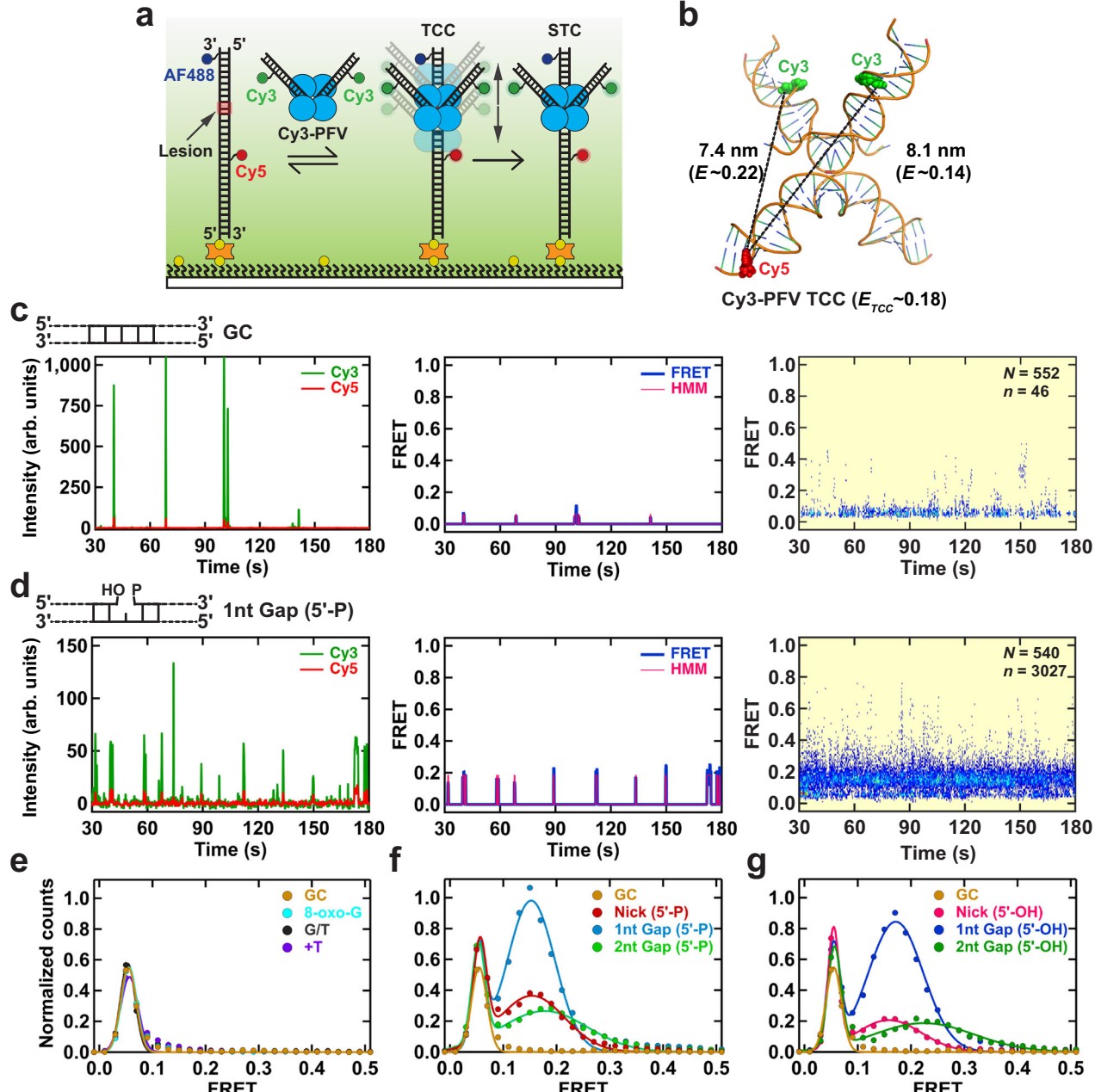

**Fig. 1 | Real-time PFV intasome target capture dynamics. a** An illustration of the smFRET experimental setup for visualizing the target capture by a PFV intasome labeled with Cy3 at non-transfer strands (Cy3-PFV). The intasome was introduced in real-time onto DNA targets containing AlexaFluor488 (AF488) and Cy5. **b** Positioning of the DNAs within the PFV TCC structure (PDB: https://doi.org/10.2210/pdb3os2/pdb) showing the fluorophore positions on the vDNAs (Cy3-PFV) and the target DNA (F-Cy5). The estimated inter-dye distances, corresponding FRET efficiencies, and the average FRET value ($E_{TCC}$) of a mixture of molecules containing a single Cy3 on the left or right vDNA. A representative intensity trajectory (left), the corresponding FRET trajectory with the HMM fit (middle), and the post-synchronized histogram (right) displaying intasome interactions with GC duplex

target DNA (**c**), and target DNA containing a 1nt Gap (5'-P) (**d**). Inset shows the total number of DNA molecules (*N*) analyzed for each substrate and the total number of transitions with >0.1 FRET (*n*). **e** Normalized smFRET histograms and Gaussian fits using 100 ms frame rate pseudo-FRET data (Supplementary Table 2), showing the distributions of FRET efficiency for target DNAs containing a GC duplex, 8-oxo-G lesion, G/T mismatch, +T insertion. Normalized smFRET histograms and Gaussian fits using 100 ms frame rate data (Supplementary Table 2) showing the distributions of FRET efficiency for target DNAs containing single strand scission (Nick), a 1 nt Gap, and a 2 nt Gap with a 5'-phosphate (5'-P) (**f**) or 5'-hydroxyl (5'-OH) (**g**). The GC duplex target DNA was included for comparison. Source data are provided in the Source Data File.

Supplementary Fig. 5f, i). Less, but statistically significant binding events were recorded for DNAs containing a nick and 2 nt gap regardless of the presence of a 5'-phosphate within the lesion (see: Supplementary Fig. 5e, g, h, j). Fitting these events to a single exponential decay resulted in a $\tau_{on}$ ($1/k_{off}$) that varied from 0.3 to 0.75 s and $\tau_{off}$ ($1/k_{on}$) that varied from 7 to 24 s (Fig. 2a, b; see Supplementary Fig. 8c–h for binned data analysis and statistics). We note that the $\tau_{on}$ events for the 2 nt gap substrates approached the frame rate, which appeared to reduce the number of recorded events and in the case of the 2 nt gap (5'-OH) artificially increased the $\tau_{off}$ (Fig. 2a, b; see Supplementary Fig. 8e, h for binned data analysis and statistics). Moreover, modest variations in intasome aggregates that lead to pseudo-FRET events may result in an altered concentration of free Cy3-PFV that could also affect the $\tau_{off}$. Nevertheless, the $\tau_{on}$ appeared to generally reflect the normalized target DNA binding efficacy.

Blocking the open end of the 1nt gap (5'-OH) DNA substrate designed to trap intasomes diffusing along the DNA[46], did not sig-nificantly alter the distribution of events per molecule or the post-synchronized FRET histogram (Supplementary Fig. 9a, b). However, we observed a significant increase in the number of normalized TCC lesion binding events (Fig. 2c). These results are consistent with the hypothesis that Cy3-PFV intasomes may bind duplex DNA non-specifically, slide along the backbone as previously shown[24], and dissociate off an open end of the target DNA before forming a stable TCC. Trapping these non-specific associations on the DNA with blocked ends increases the probability of productive TCC events. As expected, specific binding events by these trapped complexes displayed a near identical FRET efficiency compared to unblocked target DNA ($E_{Blocked\text{-}End\ 1nt\ Gap\ (5'\text{-}OH)} \sim 0.18 \pm 0.05$; see Table 1 for comparisons).

**Intasome binding increases strand transfer into the target DNA**

We observed numerous prolonged FRET events that appear consistent with the formation of a stable STC that resulted from integration of the Cy3-PFV intasomes near the lesions on the target DNA (Fig. 3a, b). Many of these STC FRET events terminated with photobleaching of the intasome or target DNA fluorophores (Fig. 3c, arrows). Post-synchronized histograms were constructed to determine the FRET efficiency ($E_{STC}$) and frequency (%STC) of STC events (Tables 1 and 2, respectively). We noted that the kinetic accumulation of STC events appeared delayed compared to TCC events (compare Fig. 1d, right with Fig. 3d; and Supplementary Fig. 4g with Fig. 3e). Moreover, target DNA substrates which resulted in frequent STC FRET events largely fit a Gaussian distribution (Fig. 3d, e, right panel; see Supplementary Fig. 10 for additional target DNA substrates), where the $E_{STC}$ generally correlated with the TCC FRET efficiency ($E_{TCC}$; see Table 1 for comparisons). Where it differed, the number of events were extremely low ($n < 5$) making interpretation of the histograms impractical (see Supplementary Fig. 10a–c). When the number of events was sufficient for interpretation, we found the average strand transfer time ($\bar{t}_{ST}$) in post-synchronized histograms did not differ significantly between the various target DNA substrates (Supplementary Table 4). The delayed-kinetics and uniformity of $E_{TCC}$ and $E_{STC}$ is consistent with the conclusion that increased PFV intasome binding to a target DNA containing nicks and gaps contributes to the formation of an STC product.

To confirm that the STC events formed a covalent integration structure we assembled PFV intasomes with viral donor oligonucleotides containing a terminal di-deoxy-adenosine (ddA; see Table 1 for Cy3-PFV-ddA DNA substrate). A ddA at the recessed 3'-vDNA end does not possess a hydroxyl-group to complete the $S_N2$ reaction that is essential for strand transfer[4]. We examined the 1 nt gap (5'-OH) target

**Table 1 | FRET efficiency of target capture complex and strand transfer complex**

| Target DNA | $E_{TCC} \pm \sigma_{TCC}$ | $E_{STC} \pm \sigma_{STC}$ |
|---|---|---|
| GC | – | 0.54 ± 0.06 |
| G/T | – | 0.32 ± 0.08 |
| Nick (5'-P) | 0.15 ± 0.08 | 0.27 ± 0.04, 0.10 ± 0.04 |
| 1nt Gap (5'-P) | 0.16 ± 0.05 | 0.16 ± 0.05 |
| 2nt Gap (5'-P) | 0.18 ± 0.10 | 0.21 ± 0.05 |
| Nick (5'-OH) | 0.17 ± 0.07 | 0.17 ± 0.05 |
| 1nt Gap (5'-OH) | 0.19 ± 0.05 | 0.18 ± 0.05 |
| 2nt Gap (5'-OH) | 0.23 ± 0.11 | 0.24 ± 0.05 |
| Cy3-PFV-ddA | | |
| 1nt Gap (5'-OH) | 0.15 ± 0.05 | - |
| Blocked-end Target DNA | | |
| 1nt Gap (5'-OH) | 0.18 ± 0.05 | 0.19 ± 0.05 |
| R-Cy5 Target DNA | | |
| 1nt Gap (5'-OH) | 0.21 ± 0.10 | 0.29 ± 0.06 |

*Wild type* Cy3-PFV or Cy3-PFV-ddA intasome interaction with Target DNA substrates labeled with Cy5 and Alexa488 (see: Supplementary Table 1). FRET values were calculated as described (*Methods*) by fitting Target Capture Complex (TCC) and Strand Transfer Complex (STC) FRET distributions with a single or two [Nick (5'P) $E_{STC}$] Gaussian distributions. *E* and *σ* indicate the mean and standard deviation, respectively.

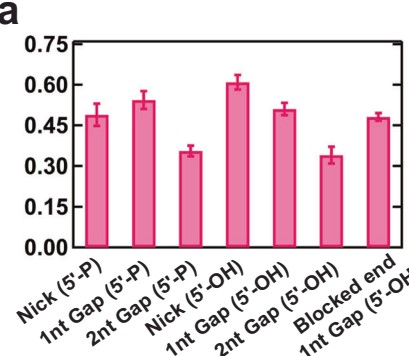

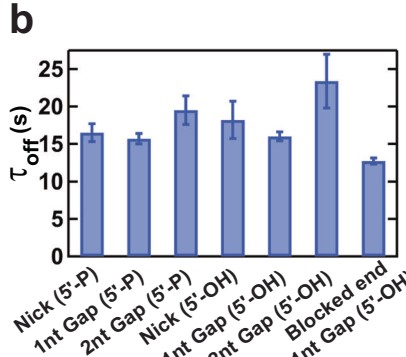

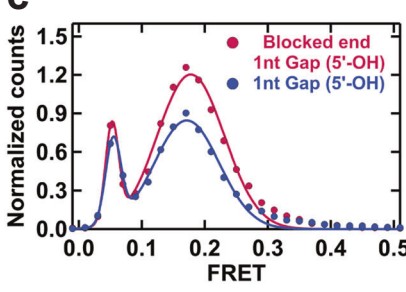

**Fig. 2 | Binding and Dissociation Lifetimes of the Target Capture Complex Events. a** The TCC binding lifetime ($\tau_{on}$) and (**b**) lifetime of the dissociated state ($\tau_{off}$) for different target DNA substrates. Bars represent the mean, and error bars represent the standard deviation from the fittings of dwell time distributions (Supplementary Fig. 8). **c** Normalized smFRET histograms and Gaussian fits (Supplementary Tables 3 and 4) showing the FRET distribution for the 1nt Gap (5'-OH) DNA with or without blocking the open DNA end. The binned data, single exponential decay fits, and number of molecules (N) for each substrate DNA examined are shown in Supplementary Fig. 8. Source data are provided in the Source Data File.

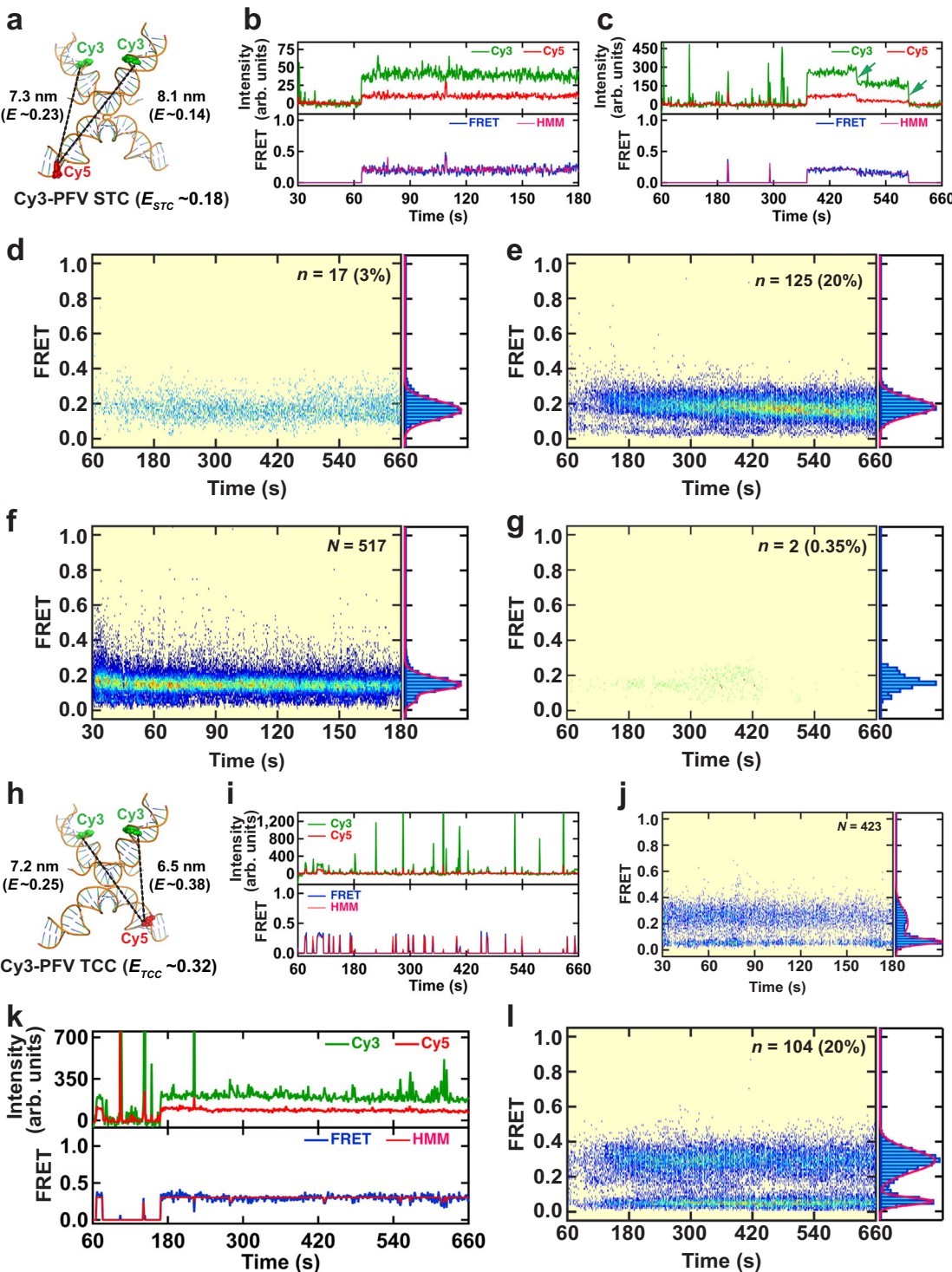

DNA as a model since it displayed the most frequent TCC and STC events (see Supplementary Fig. 5 for relative frequency of TCC and Table 2 for STC frequency across the model target DNAs). As expected, we observed significant numbers of TCC events with Cy3-PFV-ddA intasomes that displayed a similar FRET efficiency compared to *wild type* Cy3-PFV intasome interactions (compare Fig. 3f to Supplementary Fig. 4g). However, only two potential STC events were recorded with Cy3-PFV-ddA intasomes ($N = 564$; $STC_{Int\ Gap\ (5'-OH)-ddA} = 0.35 \pm 0.2\%$) compared to 20% with *wild type* Cy3-PFV intasomes (Fig. 3g; Table 2). This >50-fold reduction in STC events with Cy3-PFV-ddA intasomes strongly suggests that the prolonged FRET events displayed by the *wild type* Cy3-PFV intasomes represent covalent STC products.

We observed a similar STC FRET efficiency (Table 1), but a ~20% increase in STC frequency when blocked-end 1 nt gap (5'-OH) was included as a target DNA (Table 2, Supplementary Fig. 9e). These observations mimic the increased frequency of TCC events observed with the 1 nt gap (5'-OH) blocked-end target DNA (Fig. 2c) and are consistent with the conclusion that the increased TCC events contributes to an increased frequency of STC events.

To determine whether the positioning of the fluorophore on the target DNA influences Cy3-PFV intasome interactions, we moved the Cy5 on the undamaged strand 11 bp to the 3'-side relative to the 1 nt Gap (5'-OH) [Fig. 3h; Supplementary Table 1, R-Cy5 1 nt Gap (5'-OH)]. Similar Cy3-PFV TCC FRET events (Fig. 3i, j) and STC FRET events

**Fig. 3 | Real-time PFV intasome catalyzed strand transfer. a** Illustration of the DNA configuration found in the PFV strand transfer complex (STC) crystal structure (PDB 3OS0) showing the fluorophore positions on the vDNAs (Cy3-PFV) and the target DNA (F-Cy5). The estimated inter-dye distances, corresponding FRET efficiencies, and the average FRET value ($E_{STC}$) of a mixture of molecules containing a single Cy3 on the left or right vDNA. **b, c** Representative intensity trajectories and corresponding FRET trajectories with HMM fits showing stable strand transfer by a Cy3-PFV intasome. Green arrows mark the photobleaching of the Cy3 FRET donor. The Post-synchronized histogram and smFRET histograms corresponding to STC events for target DNAs containing a 1nt Gap (5′-P) (**d**) or 1nt Gap (5′-OH) (**e**). The Gaussian fits to the FRET histograms are shown as red lines. Data were collected at 100 ms frame rate (**b**) or 1 s frame rate (**c–e**). Inset shows the total number of transitions with >0.1 FRET (**n**) and the percentage (%) of strand transfer events. The post-synchronized histogram and smFRET histograms corresponding to TCC (**f**) and STC (**g**) events for Cy3-PFV-ddA interacting with 1nt Gap (5′-OH) target DNA. Data were collected at 100 ms frame rate (**f**) and 1 s frame rate (**g**). Inset shows the

total number of DNA molecules analyzed (**N**) (**f**); and the number of stable transitions with >0.1 FRET (**n**) and the percentage (%) of STC events (**g**; $N = 564$). **h** Illustration of the DNA configuration found in the PFV strand transfer complex (STC) crystal structure (PDB 3OS0) showing the fluorophore positions on the vDNAs (Cy3) and the reverse-Cy5 (R-Cy5) target DNA, respectively. The estimated inter-dye distances, corresponding FRET efficiencies, and the average FRET value ($E_{TCC}$) of a mixture of molecules containing a single Cy3 on the left or right vDNA. **i** A representative intensity trajectory (top) and the corresponding FRET trajectory (bottom) with the HMM fit showing Cy3-PFV binding to R-Cy5 target DNA containing a 1nt Gap (5′-OH). **j** The post-synchronized histogram (left) and the smFRET histogram (right) produced by averaging the total number (**N**) of TCC smFRET traces. **k** A representative intensity trajectory (top) and the corresponding FRET trajectory (bottom) with HMM fit showing Cy3-PFV strand transfer into R-Cy5 target DNA containing a 1nt Gap (5′-OH). **l** The post-synchronized histogram (left) and the smFRET histogram (right) corresponding to the total number (**n**) and percentage (%) of STC events. The data in (**i**–**l**) were collected at 1 s frame rate.

(Fig. 3k, l) were observed with the R-Cy5 1 nt Gap (5′-OH) target DNA compared to the original 1 nt gap (5′-OH) target DNA containing the Cy5 fluorophore on the undamaged strand 11 bp to the 5′-side of the lesion (compare Fig. 3j to Supplementary Fig. 4g; and Fig. 3l to Fig. 3e). We conclude that the fluorophore location has little or no effect on the TCC or STC events. Intriguingly, the FRET efficiency of the TCC and

STC events with the R-Cy5 1 nt gap (5′-OH) was greater than the FRET efficiency of the TCC and STC events with the Cy5 located symmetrically on the other side of the 1 nt gap (5′-OH). These results are consistent with the conclusion that the Cy3-PFV intasome binds asymmetrically on the lesion-containing target DNA, with a preferred binding to the 3′-side of the 1 nt gap (5′-OH).

## DNA breaks and gaps target site-specific PFV integration

To map the STC events we examined strand transfer products by denaturing gel electrophoresis (Gel; Fig. 4). We first examined unlabeled PFV intasomes incubated with the target DNAs utilized in the smFRET analysis (Supplementary Table 1). No changes in the substrate were observed in negative control reactions (-PFV) when fully duplex target DNA (G/C) as well as target DNA containing an 8-oxo-G, G/T mismatch (G/T) or a +T insertion (+T) were examined (Fig. 4b). However, additional bands of varying intensity were consistently observed with a target DNA containing a nick (5′-P or 5′-OH), a 1 nt gap (5′-P or 5′-OH) or a 2 nt gap (5′-P or 5′-OH) that are consistent with PFV integration products (Fig. 4b). The relative pattern of STC events observed by smFRET generally mimicked the Gel analysis (Fig. 4a), with differences in the absolute integration frequency attributable to the 5-fold lower concentration of Cy3-PFV intasomes utilized in smFRET that is necessary to moderate background fluorescence (5 nM for smFRET compared to 25 nM for gel analysis).

Most strand transfer events observed by Gel analysis were the result of half-site integration, where only one vDNA is covalently joined to the target DNA (Fig. 4b). For example, the target DNA containing a 1 nt gap, nick and 2 nt gap with a 5′-P resulted in major Cy5-DNA bands of 82, 81, and 80 nt, respectively (Fig. 4b, left). These correspond to strand transfer of a 38 bp vDNA into the undamaged Cy5-containing strand 44, 43, 42 nt from the 3′-end, respectively (Fig. 4c). Because the two PFV strand transfer events are normally separated by 4 bp[23,31], the position of the second vDNA strand transfer event would be located on the lesion-containing strand within the 1 nt gap, at the nick, or at the 5′-side of the first missing nucleotide within the 2 nt gap. In all these cases there is no phosphate bond to complete the isoenergetic strand transfer chemistry. A similar integration pattern was detected with target DNAs containing a 1 nt gap, nick and 2 nt gap with a 5′-OH (Fig. 4b, right), except that the 2 nt gap (5′-OH) contains equivalent intensity bands at 80 and 82 nt (Fig. 4b, right). These results suggest that in the absence of a 5′-P, the PFV intasome positions the chemically inert non-transferred vDNA strand between the 3′-OH and the first missing nucleotide or the 3′-OH and the second missing nucleotide on the lesion-containing strand. The asymmetric location of these integration products parallels the asymmetric binding of Cy5-PFV intasomes 3′ of the lesion predicted by TCC and STC FRET events as well as the broadening and increased FRET efficiency of TCC events observed by smFRET with target DNA containing a 2 nt gap (5′-OH). Together

## Table 2 | Frequency (%)[a] of strand transfer complex events

| Cy3-PFV | | | PFV |
|---|---|---|---|
| **Target DNA** | **%STC$_{smFRET}$ ± σ %STC$_{smFRET}$** | **Number of DNA molecules (N)** | **%STC$_{Gel}$ ± σ%STC$_{Gel}$** |
| GC | 0.3 ± 0.6 | 585 | 0.4 ± 0.6 |
| 8-oxo-G | 0 ± 0 | 624 | 0.9 ± 1.5 |
| G/T | 0.5 ± 0.7 | 616 | 1.4 ± 0.8 |
| +T | 0 ± 0 | 531 | 0.8 ± 0.7 |
| Nick (5′-P) | 0.7 ± 0.8 | 587 | 3.2 ± 2.9 |
| 1nt Gap (5′-P) | 3 ± 1.7 | 559 | 8.4 ± 3.8 |
| 2nt Gap (5′-P) | 1.4 ± 1.2 | 506 | 10.7 ± 5.7 |
| Nick (5′-OH) | 1.9 ± 1.4 | 626 | 20 ± 12.4 |
| 1nt Gap (5′-OH) | 20.4 ± 4.0 | 614 | 29.3 ± 10.9 |
| 2nt Gap (5′-OH) | 11.7 ± 3.2 | 605 | 14.2 ± 1.7 |
| **Cy3-PFV-ddA** | | | |
| **Target DNA** | **%STC$_{smFRET}$ ± σ %STC$_{smFRET}$** | **Number of DNA molecules (N)** | |
| 1nt Gap (5′-OH) | 0.4 ± 0.6 | 564 | |
| **Blocked-end Target DNA[b]** | **%STC$_{smFRET}$ ± σ %STC$_{smFRET}$** | **Number of DNA molecules (N)** | |
| 1nt Gap (5′-OH) | 24.5 ± 4.3 | 547 | |
| **R-Cy5 Target DNA[b]** | **%STC$_{smFRET}$ ± σ %STC$_{smFRET}$** | **Number of DNA molecules (N)** | |
| 1nt Gap (5′-OH) | 19.7 ± 4.0 | 527 | |
| **Cy3/Cy5-PFV** | | | |
| **Unlabeled Target DNA** | **%STC$_{smFRET}$ ± σ %STC$_{smFRET}$** | **Number of DNA molecules (N)** | |
| GC | 0.4 ± 0.6 | 548 | |
| 1nt Gap (5′-OH) | 15.6 ± 3.6[c] | 609 | |

[a]Percentages were calculated as described (*Methods*). *N* indicates the total number of DNA molecules analyzed for each experiment. *n* is the number of molecules that showed strand transfer events. Source data are provided in the Source Data File.
[b]Integration events with Cy3-PFV intasomes.
[c]Includes PFV intasomes containing Cy5-only integration products. Integration by PFV intasomes containing Cy3-only would not be detected in this smFRET system and could theoretically introduce an additional 5% STC, which would be virtually identical to 1 nt Gap (5′-OH) STC events (above).

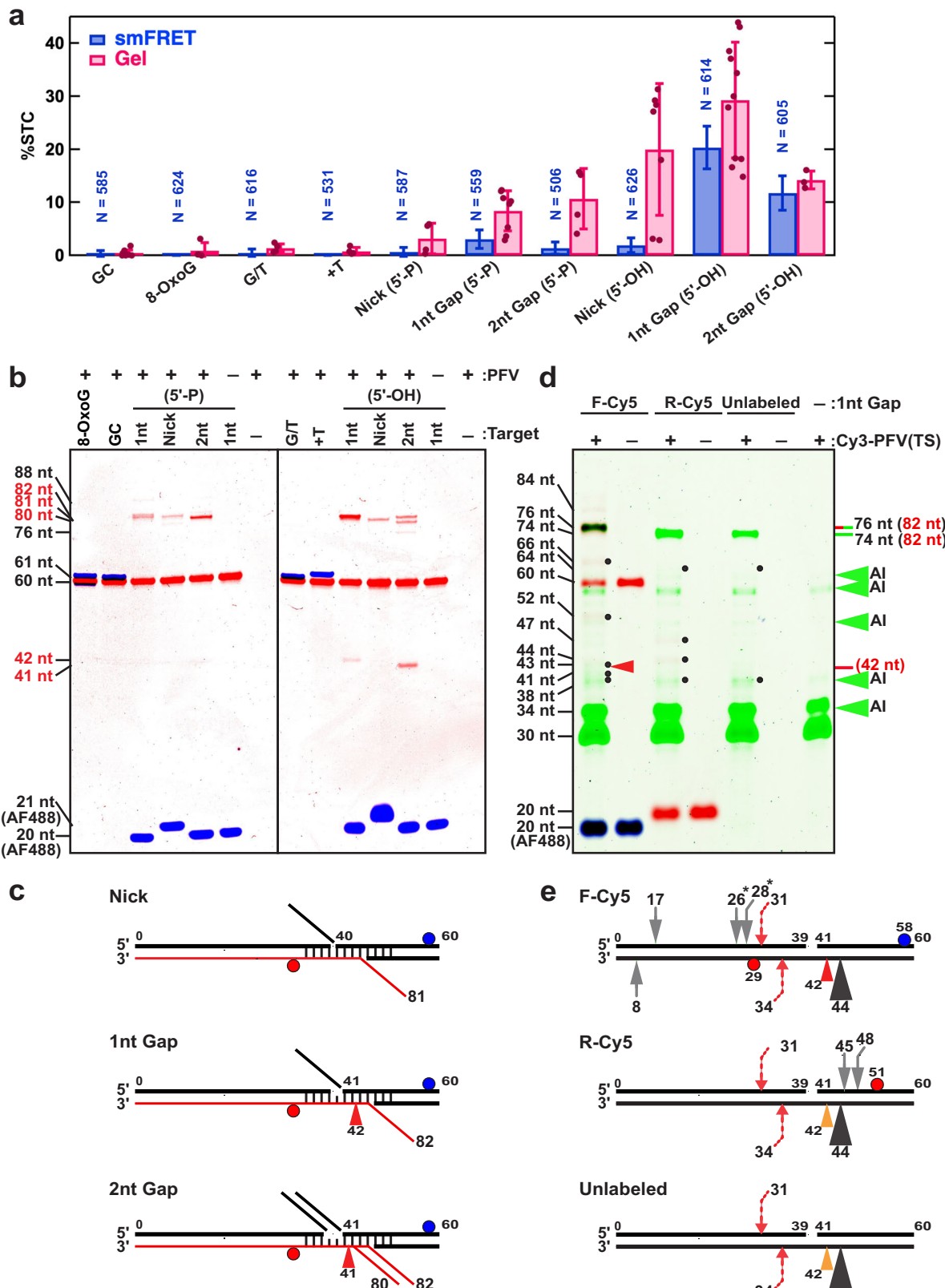

these results appear to imply that the presence of a 5′-P influences the location of half-site strand transfer events by altering the stable positioning of PFV intasomes.

We also detected 42 nt and 41 nt DNA products with the 1 nt gap (5′-OH) and 2 nt gap (5′-OH), respectively (Fig. 4b, right; Fig. 4c). These DNA products appear consistent with an alcoholysis activity previously

reported for monomeric HIV-1 IN, Maedi Visna Virus IN, and Rous Sarcoma Virus IN[47–50]. Our results provide the first observation of alcoholysis by an intasome or by PFV IN. Alcoholysis is dependent on the presence of a nucleophile, such as glycerol that is present in our storage and reaction buffers, to nick viral or target DNA. While IN mediated alcoholysis of viral DNA appeared to be site specific, this

**Fig. 4 | Analysis of PFV strand transfer activity. a** Quantification of single-molecule FRET (smFRET) and denaturing gel electrophoresis (Gel) analysis of STC integration activity (%) on different DNA targets. Bars indicate the mean and error bars the standard of deviation of smFRET events that were observed on N molecules or from at least three independent Gel quantification analysis (see: •). The smFRET data were generated using Cy3-PFV, and the gel data were obtained using unlabeled PFV with Cy5 and AlexaFluor488 (AF488) labeled target DNAs. Error bars in the gel data are the standard deviations from triplicates. Differences in absolute frequency reflect different intasome reaction concentrations in smFRET (5 nM) and gel analysis (25 nM). **b** Representative denaturing PAGE gels from bulk integration studies. The lengths of ssDNAs derived from a Sanger sequencing ladder are shown. The blue and red bands show DNA fragments containing AF488 and Cy5, respectively. The target DNA substrates used for each lane are shown above (Supplementary Table 1). **c** Schematics showing the major strand transfer event with predicted ssDNA length (red line) and alcoholysis (red arrowhead) exhibited by unlabeled PFV intasomes. **d** Gel analysis (top) and integration sites (bottom) of PFV

intasomes labeled with Cy3 on the vDNA transferred strand [Cy3-PFV (TS)]. Integration into different 1nt Gap (5′-OH) substrates containing a forward Cy5 label (F-Cy5), reverse Cy5 label (R-Cy5) or unlabeled as illustrated below. The blue, green and red color gel bands correspond to DNA fragments containing AF488, Cy3, and/or Cy5. The brown color bands contain both Cy3 and Cy5. Green arrows indicate location of autointegration (AI) products. The black dots indicate products resulting from strand transfer and red arrowhead the 42 nt alcoholysis product (see: **b,c**). Orange arrowhead in illustration represents the predicted location of an undetected alcoholysis product since the DNA strand does not contain a fluorophore label. **e** Calculated integration sites for each target DNA. Large black arrow indicates major (>90%) half-site integration product similar to major bands in Panel b; red dashed arrows indicate minor concerted strand transfer product; gray arrows show minor half-site products. The numbers indicate the positions of integration relative to the 5′-end of the top target DNA strand. * Indicates products that have more than one integration site solution (see text). Source data are provided in the Source Data File.

---

endonucleolytic activity has historically appeared to be non-specific leading to strand breaks throughout a target DNA[47–50]. However, we only observed the alcoholysis break on the undamaged strand of the target DNA, suggesting PFV intasomes that are stalled at a break/gap lesion can promote significant site-specific endonucleolytic alcoholysis.

A weak but clearly visible DNA product of ~76 nt was observed with the 2 nt gap (5′-OH) target DNA (Fig. 4b). This product could conceivably be a concerted integration event since the second strand transfer should occur 35 nt from the 5′-end on the lesion-containing strand, which would be undetectable with unlabeled PFV intasomes. To determine whether additional integration events occur along the target DNA, we assembled PFV intasomes containing Cy3-labeled vDNA on the transferred strand [Cy3-PFV (TS); see Supplementary Table 1]. Cy3-PFV (TS) integration events transfer a 30 nt vDNA to an unlabeled target DNA (unlabeled), a target DNA containing a Cy5-fluorophore located on the undamaged strand on the 5′ side relative to the lesion (F-Cy5; see Supplementary Table 1), or a target DNA containing a Cy5-fluorophore located 3′ on the lesion-containing strand (R-Cy5; see Supplementary Table 1). We examined the 1 nt gap (5′-OH) (Fig. 4d; Supplementary Fig. 11a) and 2 nt gap (5′-OH) (Supplementary Fig. 11b) target DNA substrates since they initially revealed integration products other than the major half-site strand transfer events (see Fig. 4b). The principal Cy3-PFV (TS) integration events (>90%) produced fragments of 76 nt or 74 nt for both the 1 nt Gap (5′-OH) and 2 nt Gap (5′-OH) (Fig. 4d; Supplementary Fig. 11a, b). Subtracting the 30 nt vDNA from the strand transfer product and accounting for a 2 nt apparent size increase when both a Cy3 and Cy5 fluorophore are present, the calculated integration events occurred at 44 nt on the undamaged strand (Fig. 4d; Supplementary Fig. 11a, b; black arrowhead). This is the identical location that produced the 80 nt and 82 nt half-site integration products with the unlabeled PFV intasomes (Fig. 4b). We also detected background levels of the 41 nt and 42 nt alcoholysis products with the F-Cy5 target DNA substrates (Fig. 4d; Supplementary Fig. 11a, b; red arrowhead) that would be undetectable with the R-Cy5 target DNA and PFV intasomes containing unlabeled vDNAs (Fig. 3d; Supplementary Fig. 11; orange arrowhead).

Minor Cy3-PFV (TS) integration products were identified with 1 nt gap (5′-OH) and 2 nt gap (5′-OH) target DNA substrates (Fig. 4d, black dots), that could be enhanced by increasing the image contrast (Supplementary Fig. 11a, b). These additional integration events accounted for ≤5% of the integration products, with the majority localized around the lesion (gray arrows) and including a distinct concerted integration product adjacent to the lesion (red dashed arrow; Supplementary Fig. 11a, b, bottom). Half-site integration products mapped to 26 nt and 28 nt on the F-Cy5 target DNA could be symmetrically localized to the non-lesion containing strand at 11 nt and 13 nt, respectively, and are therefore marked with a star (Supplementary

Fig. 11a, b, bottom). The 26, 28, 45 and 48 integration products appeared to be equidistant from the Cy5 fluorophore location (Supplementary Fig. 11a, b, bottom), suggesting fluorophore DNA lesions might contribute to minor retroviral integration products. More products could be visualized with increased image contrast but accounted for even less frequent events. Together, these results confirm the conclusion that nicked or gapped significantly enhances integration events, both half-site and concerted, at or near the site of the lesion.

## Structural dynamics during target DNA binding and integration by PFV intasomes

Structural studies suggest that the spatial positioning of the vDNA within the TCC and STC are nearly identical, with similar calculated FRET efficiency (Fig. 5b)[23,31]. However, the dynamic processes associated with the transition between TCC and STC has not been explored. To address these questions, we assembled PFV intasomes with two donor vDNAs, one containing a Cy3 fluorophore and the other containing a Cy5 fluorophore (Cy3/Cy5-PFV; Fig. 5a, b; *Methods*). FRET changes associated with alterations in the relative geometry of the two vDNAs within the Cy3/Cy5-PFV intasomes were then monitored in real-time during TCC and STC formation with the model 1 nt gap (5′-OH) target DNA.

In these studies, we first localized the 1 nt gap (5′-OH) target DNAs in the smFRET flow cell and then photobleached the Cy5 fluorophore prior to injection of Cy3/Cy5-PFV intasomes (Fig. 5a, red). We maintained the Alexa488 in the target DNA both for consistency with previous substrates and as an additional fluorophore that might be utilized for localization. Transient Cy3/Cy5-PFV binding events were observed that appeared similar to Cy3-PFV TCC interactions (Fig. 5c, 1 s frame rate; Supplementary Fig. 12e, 100 ms frame rate; see Fig. 1d and Supplementary Fig. 4g for comparisons). A significant number of lower FRET tails appeared to extend from the higher FRET TCC interactions at both 1 s (Fig. 5c) and 100 ms (Supplementary Fig. 12e) frame rates. In contrast, the formation of stable STC's by Cy3/Cy5-PFV intasomes on the 1 nt gap (5′-OH) target DNA resulted in a relatively narrow distribution of higher FRET events with few trailing lower FRET events (Fig. 5d). Taken together, these observations appeared to suggest that the increased trailing TCC events might result from more dynamic vDNA motions compared to STC events. However, they could also reflect time-averaged FRET events by TCC Cy3/Cy5-PFV intasome. We plotted the distribution and determined the FRET efficiency of TCC ($E_{Cy3/Cy5-PFV} = 0.62 \pm 0.1$, mean ± σ) and STC ($E_{Cy3/Cy5-PFV} = 0.66 \pm 0.09$, mean ± σ) events at 100 ms frame rate (Supplementary Fig. 12f). While the mean and width of gaussian fits was not statistically different, the tail of lower FRET TCC events remained evident in an overlay of these plots (Supplementary Fig. 12f). These results continue to infer that the TCC vDNA may be more dynamic than STC vDNA, however further studies will be required to fully resolve this issue.

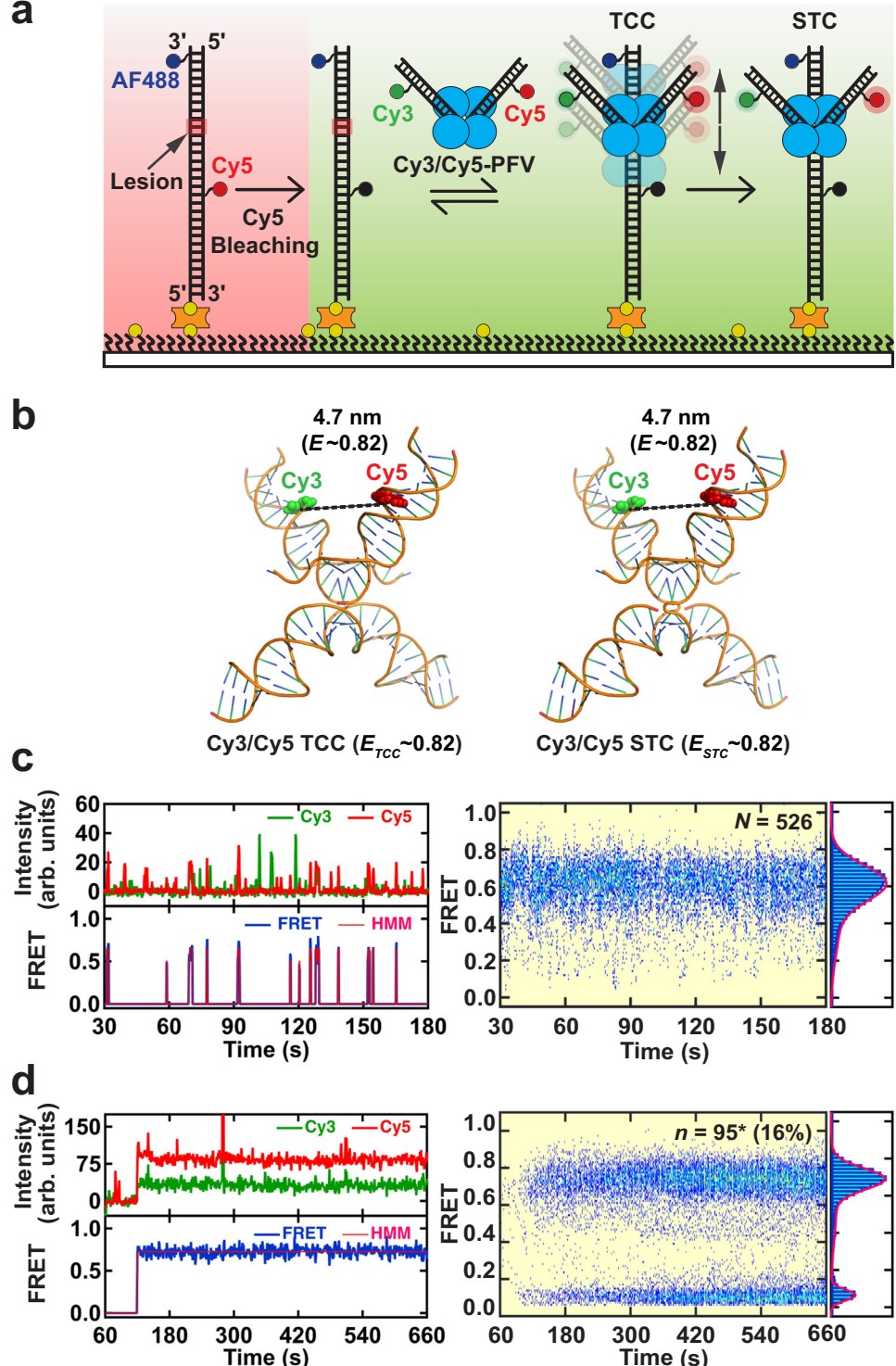

**Fig. 5 | Probing the structural dynamics of PFV intasome during target capture and strand transfer. a** An illustration of the smFRET experimental setup for visualizing the structural dynamics of PFV intasomes during TCC and STC formation. A PFV intasome labeled with Cy3 and Cy5 at the non-transfer strands (Cy3/Cy5 PFV) was introduced after photobleaching Cy5 on the substrates. **b** Illustration of the DNA configuration found in the PFV TCC structure (PDB 3OS2) and STC structure (PDB 3OS0) with the fluorophore positions on the vDNAs (Cy3 and Cy5). The estimated inter-dye distances, corresponding FRET efficiencies, and the average FRET value ($E_{TCC}$ and $E_{STC}$) are shown. **c** Left: A representative TCC intensity trajectory (top) and the resulting FRET trajectory (bottom) with the HMM fit showing Cy3/Cy5-PFV binding to a 1nt Gap (5′-OH) target DNA (left). Right: Post-

synchronized histogram (left) and smFRET histogram (right) generated by averaging the total number ($N$) of TCC FRET traces. **d** Left: A representative STC intensity trajectory (top) and the resulting FRET trajectory (bottom) with HMM fit showing Cy3/Cy5-PFV integration into a 1nt Gap (5′-OH) target DNA (left). Right: Post-synchronized histogram (left) and smFRET histogram (right) generated by averaging the total number ($n$) of STC Cy3/Cy5-PFV FRET traces (* indicates Cy3/Cy3-PFV events are included in the total number of STC traces). The percentage (%) is the efficiency of Cy3/Cy5-PFV strand transfer that includes Cy3/Cy3-PFV bleed-through events (6%). The Gaussian fits to the histograms are shown as red lines. The TCC and STC data were collected at 1 s frame rate.

There was also a significant peak of very low pseudo-FRET events that appeared when the studies were performed at a 1 s frame rate (6% of the products). When examined individually these pseudo-FRET events were found to result from STCs formed by intasomes with two Cy3 labeled vDNAs (Cy3/Cy3-PFV). Excitation of these Cy3/Cy3-PFV STC events resulted in increased emission intensity that bled through into the Cy5 channel resulting in pseudo-FRET. Including the calculated the frequency of Cy5/Cy5-PFV intasome STC events that are undetectable with this smFRET system (~5%) to the total number of STC events, the total frequency of STC events with these PFV intasomes (~21%) appeared nearly equivalent to the frequency of STC events observed with the Cy3-PFV intasomes (20%; see Table 2). Overall, these observations suggest that the vDNA within PVF intasomes may be slightly more dynamic during TCC target DNA interactions compared to the relatively stable vDNA geometry of a STC on the target DNA. Since four-way DNA is dynamic even in the absence of nicks or gaps at the junction that should further increase mobility[51], these observations suggest STC formation may result in a relatively stable integrase complex that remains associated with the covalent vDNA-target DNA structure.

## Discussion

Previous single molecule imaging of PFV intasomes demonstrated that site recognition limits integration[24]. Once a DNA site is identified the two strand transfer steps that result in concerted integration occur very fast (< 0.5 s)[24]. Historical biochemical and structural studies have suggested that retroviruses prefer bent and/or flexible DNA at the integration site[21,23,25,52]. Here we show that intrinsically flexible but intact duplex DNA containing a G/T mismatch or a +T nucleotide insertion that induces a stable DNA bend[41–43,53], does not significantly enhance PFV intasome binding or integration. The DNA substrates that have suggested a preference for bent and/or flexible DNA also incorporate altered the duplex DNA twist. In addition, PFV intasomes prefer supercoiled target DNAs[24] that undergo continuous and dynamic interchange between plectonemic supercoils and untwisted duplex DNA[54]. PFV intasome STC structures in the presence of naked or nucleosome-bound target DNA show untwisted DNA targets, where strand transfer appears to occur across a backbone that averages ~12 bp per turn[23,25]. Taken together, these observations appear consistent with a hypothesis that PFV intasomes recognize and integrate into dynamic untwisted target DNA sites where the backbone may be more flexible and/or malleable[53]. Additional studies will be necessary to fully interrogate the DNA conformation(s) that facilitate stable binding geometry and retroviral integration.

A significant increase in PFV intasome binding was detected when the target DNAs contained DNA strand breaks and gaps. These observations appear to support the notion that an intrinsically flexible DNA backbone can act as a binding sink. Importantly, the increased number of intasome-DNA interactions correlated with increased site-specific strand transfer events at the break/gap site. This appears to be the first targeted strand transfer events observed for any retrovirus. Analysis of the TCC smFRET binding events suggested that the PFV intasome asymmetrically localized to the 3′-side of the break/gap. Integration site analysis correspondingly demonstrated asymmetric strand transfer to the 3′-side of the lesion. These results are consistent with a model where the PFV intasome stalls on the damaged DNA strand 3′ of the missing nucleotide (or strand break), prompting the typical integrase-mediated strand transfer on the undamaged DNA strand four bp from the break/gap. Concerted strand transfer into the lesion-containing strand would not be possible because the phosphate bond required for the isoenergetic chemistry is missing, resulting in half-site integration as the most common product (Fig. 4).

Assuming similar mechanics, these observations offer a plausible explanation for the reduced HIV-1 integration in cells containing *ogg1*, *mutyh*, and *polB* BER pathway mutations, as well as loss of the G

preference at consensus HIV-1 integration strand transfer junctions[8,11]. We consider a model in which OGG1 protein removes an 8-oxo-G lesion and creates a 1-nt gap that results in enhanced HIV-1 intasome TCC interactions. Associated integration events would principally result in half-site products and a DNA double strand break (DSB) that is processed by DSB repair[55]. Both homology-dependent recombination (HR) and non-homologous end-joining (NHEJ) DSB repair would most likely excise some if not all the extraneous half-site retroviral DNA leading to largely non-productive cellular infections.

However, increased intasome-lesion TCC interactions also enhanced less frequent integrations events (Fig. 4). For example, we found that as much as 5% of the integration events target adjacent sites surrounding the nick/gap. BER pathway mutations that are unable to create the nick/gap lesion would be predicted to reduce or eliminate these integration events, ultimately decreasing the frequency of productive HIV-1 infection. We note an intriguing possibility that relies on the observation that the MUTYH protein interacts with abasic sites shielding them from APE/Ref-1 and preserving the phosphate bond following glycosylase removal of damage[56]. In this case, a combination of OGG1 and MUTYH activities might create a DNA lesion that supports additional concerted integration events rescuing them from DSB formation, and accounting for the genetic effect of both *ogg1* and *mutyh* in HIV-1 integration. The loss of the 5′-G preference within consensus HIV-1 integration sites observed in *ogg1* mutant cells would reflect the loss of integration events that target 8-oxo-G lesions processed by OGG1.

A modest but consistent increase in vDNA conformational dynamics was apparent when Cy3/Cy5-PFV intasome TCC events were compared to STC events (Fig. 5). Diminished STC conformational dynamics could indicate that the complex remains associated with the target DNA following integration (Fig. 5). Single molecule magnetic tweezer studies by our group showed that under very low force (~0.1 pN), any bound PFV integration complex permits free rotation of DNA strands around their common axis that releases supercoils as well as dissociation of a DSB produced by concerted strand transfer of short vDNA oligonucleotides[24]. However, another group seemed to suggest that a PFV intasome-STC complex may be capable of tethering the DSB ends even at high force (>30 pN)[30]. We note that these studies employed atypical buffer conditions where PFV intasomes may undergo partial or complete loss of activity as well as significant aggregation[57]. Even under our well-defined conditions, we detected sporadic aggregates as background pseudo-FRET excursion events (see: Fig. 1; Supplementary Movie 1 for examples). We regard it plausible that PFV integration events displaying high force stability might reflect unusual intasome aggregates bound to the target DNA or half-site products generated from partially active intasomes. Such abnormal products might result in uncommon tethers that are only susceptible to high-force mechanical breakage.

Retroviral vectors are attractive for gene therapy applications in part due to the stable integration of a transgene[58]. While lentiviral vectors derived from HIV-1 are able to infect non-dividing cells[59], they often integrate into actively transcribing genomic regions[60]. Such integration events may unintentionally alter cellular functions or activate oncogenes[61]. Modulating the target site choice of a non-pathogenic retrovirus could be one scheme for reducing the potential pathogenicity of targeted integration. The results presented here suggest that prior modification of the target DNA backbone and/or configuration could be an additional method for targeting gene therapy with retroviral vectors.

## Methods

### Preparation of target DNAs

Single strand DNA (ssDNA) DNA oligonucleotide sequences in this study are listed in Supplementary Table 1. The 8-oxo-G ssDNA was purchased from Midland Certified Reagent Company. All other ssDNAs

were purchased from Integrated DNA Technologies. ssDNAs were labeled with NSH-ester of AlexaFluor488 (AF488; GE Healthcare), sulfo-Cy3 (Lumiprobe) or sulfo-Cy5 (Lumiprobe) at C6 amino modifications following a standard labeling protocol[40]. The labeled and unlabeled DNAs were separated on a C18 column (Agilent Technologies) using reverse-phase HPLC. Selected HPLC fractions were concentrated using 0.5 mL, 3 kDa molecular weight cutoff Amicon Ultra centrifugal filters after evaporating the organic solvent in a SpeedVac vacuum concentrator (Savant). The DNAs were buffer exchanged into 20 mM Tris-HCl, pH 8.0, 1 mM EDTA, and stored at −20 °C. Each labeled ssDNA was gel purified using 12% Acrylamide:Bis 19:1, 7 M Urea PAGE. The gel purified DNAs were concentrated as above in 20 mM Tris-HCl, pH 8.0, 1 mM EDTA and stored at −20 °C.

The substrate double strand DNAs (dsDNA) in Supplementary Table 1 were obtained by annealing ssDNAs in 20 mM Tris-HCl, pH 8.0, 1 mM EDTA, 100 mM NaCl. Annealing reactions were performed in a thermal cycler (Applied Biosystems) by heating the samples to 95 °C and slowly cooling to 15 °C. Fully annealed dsDNAs were enriched by anion exchange HPLC on a Gen-Pak FAX column (Waters). The purity of HPLC fractions was assayed by 5% Acrylamide:Bis 59:1 native PAGE. Fractions containing ~100% dsDNA were pooled and concentrated as above. The dsDNA concentrations were determined by absorbance at 280 nm, AF488 absorbance at 490 nm, Cy3 absorbance at 550 nm, and Cy5 absorbance at 650 nm (NanoDrop, Thermo Fisher Scientific). DNAs were stored in 20 mM Tris-HCl, pH 8.0, 1 mM EDTA, 100 mM NaCl at −20 °C.

## PFV intasome assembly

PFV intasomes were assembled as described previously using recombinant integrase (IN) and dsDNA mimicking the PFV U5 vDNA ends (Supplementary Table 1)[24,27,62,63]. vDNAs were prepared as described above (Supplementary Table 1). The intasome assemblies were performed by salt dialysis followed by chromatographic purification on a size exclusion column (SEC, Superose 6 Increase, GE Healthcare). The integration activity of SEC fractions was quantified with supercoiled plasmid DNA target before flash freezing and storing at −80 °C. Catalytically deficient Cy3-PFV-ddA intasomes were assembled in the same way using integrase and a vDNA containing 3′-ddA (Supplementary Table 1)[23,26]. The intasome containing the Cy3 and Cy5 FRET pair (Cy3/Cy5-PFV) was prepared by mixing equimolar Cy3 vDNA and Cy5 vDNA during the assembly reaction. The expected outcomes for different species are: 50% intasomes with Cy3 vDNA and Cy5 vDNA, 25% intasomes with two Cy3 vDNA, and 25% intasomes with two Cy5 vDNA.

## smFRET imaging

Real time smFRET imaging was performed on a custom assembled inverted fluorescence microscope (Olympus), as described previously[40]. Prism-based total internal reflection fluorescence (TIRF) of a green (532 nm) or red (635 nm) lasers were used to excite fluorophore labeled target DNAs attached to a flow cell surface. The fluorescence from individual fluorophores was collected through a 60X water immersion objective (Olympus) and directed onto an emCCD chip (Princeton Instruments) after magnifying another 1.6X and separating Cy3 and Cy5 emissions using a Dual View Simultaneous Image Splitter (Photometrics).

The quartz surface of the flow cells was passivated with a 1:20 ratio of biotin-PEG and methoxy-PEG (5000 MW, Laysan Bio). Biotin-PEG was used to immobilize biotin-labeled target DNAs by biotin-neutravidin-biotin linkages at ~0.2 molecules/μm² surface density. Methoxy-PEG minimizes non-specific surface interactions[34,40]. The imaging buffer (Buffer-I) for all experiments was 30 mM Bis-tris propane, pH 7.5, 110 mM NaCl, 2 mM MgSO₄, 4 μM ZnCl₂, 0.1 mM DTT, 0.2 mg/mL BSA, 0.02% IGEPAL (Sigma). Buffer-I also included saturated (~2 mM) Trolox and an oxygen scavenging system (OSS) to minimize photo-blinking and photobleaching of fluorophores,

respectively[40]. The OSS consisted of 25 mM protocatechuic acid (PCA) and 20 nM protocatechuate dioxygenase (PCD)[40,64]. All the experiments were performed at 24 ± 1 °C.

## smFRET target capture assays

The imaging for target capture was performed at 100 ms time resolution to capture transient events. Single-molecule movies were initiated by exciting Cy5-DNA in Buffer-I with a 635 nm red laser at ~2 mW. After 20 s, the excitation was switched to a 532 nm green laser maintained at ~6 mW. 10 s after the green laser exposure, 5 nM PFV intasomes with Cy3-PFV in Buffer-I were infused to the flow cells. Data recording was continued under continuous green laser excitation for 2.5 min from the injection (Supplementary Fig. 2a).

Experiments with PFV intasomes with Cy3/Cy5-PFV were performed the same way with the following modifications. The initial red laser exposure (30 s) was used to entirely bleach Cy5 in the field of view. The fast photobleaching was achieved by eliminating OSS in Buffer-I. The subsequent intasome injection and imaging were performed with Buffer-I including OSS (Supplementary Fig. 2b).

## smFRET strand transfer assays

Strand transfer assays were recorded with 1 s time resolution and at lower laser powers to improve fluorophore lifetimes and allow longer observations. Movies were initiated by Cy5 excitation with the red laser at ~1 mW. After 30 s, the excitation was switched to the green laser maintained at ~4 mW. 30 s after the green laser exposure, 5 nM PFV intasomes with Cy3 labeled vDNA in Buffer-I were infused to the flow cell. The imaging was under continuous green laser excitation for 10 min from the injection (Supplementary Fig. 2c).

Experiments with PFV intasomes where the vDNA was labeled with both Cy3 and Cy5 were performed with the following modifications. The initial red laser exposure (30 s) was used to entirely bleach Cy5 in the field of view. The fast photobleaching was achieved by eliminating the OSS in Buffer-I. The subsequent intasome injection and imaging were performed with Buffer-I including OSS (Supplementary Fig. 2d).

The R-Cy5 1nt Gap (5′-OH) experiments displayed fast photobleaching and photophysical fluctuations of Cy5 due to its closeness to purines (A,G)[65] in the DNA sequence (Supplementary Table 1). Therefore, experiments were performed at 1 s resolution and ~2 mW and at ~4 mW for photobleaching.

## Initial processing of SM movies

Single-molecule movies were acquired using the Micro-Manager imaging software as described previously (Supplementary Movies 1 and 2)[40]. We used a custom MATLAB (MathWorks) program to extract intensity and FRET data from these movies as follows. Cy3 and Cy5 channels were mapped with the emissions of 0.2 μm crimson carboxylate modified microspheres (Thermo Fisher Scientific). These movies were manually inspected to identify single-molecules as diffraction-limited spots. Each Cy5 molecule is then marked with a circle of adjustable radius. To avoid interference of neighbor molecules, the radius was set to three-pixels to encircle one molecule and exclude neighbors. In addition, the distance between molecules or the edges of the FOV was set to five pixels. The eccentricity (a measure of circularity) was set at 0.2. This selection criteria allowed us to identify ~500–600 well separated target DNA molecules per movie.

## Generation of single molecule traces and Hidden Markov Model analysis

Pixel intensities within a given circle were integrated to generate raw Cy3 and Cy5 intensities. This procedure was continued for all movie frames to collect emissions as a function of time and to build intensity traces without background corrections (Supplementary Fig. 2a–d, Left Panel). The graphical user interface (GUI) of our MATLAB program

allowed direct comparison of these traces with their corresponding overlapped Cy3, Cy5 spots in the single-molecule movies.

The infusion of PFV intasomes with Cy3 labeled vDNA led to marked increase in both Cy3 and Cy5 backgrounds. These intensity jumps were used as references for trace truncations and background corrections (Supplementary Fig. 2a–d, Left-middle Panel). The 100 ms traces were smoothed with a three-point averaging and the 1 s traces were not smoothed. Truncated traces were subjected to a vbfret Hidden Markov Model (HMM) algorithm built into our program[39] to first adjust the backgrounds and then to identify states in FRET traces. FRET was calculated from corrected intensities (I) as $I_{Cy5}/(I_{Cy3} + I_{Cy5})$. Four states were used as the initial guess for HMM. When an intasome is not bound to a target DNA, both $I_{Cy3}$ and $I_{Cy5}$ approached zero in the background corrected intensity traces. This led to erratic fluctuations in FRET for intasome unbound regions (Supplementary Fig. 2a–d, Right-middle Panel). As a solution, the FRET and the corresponding HMM state was assigned zero when either $I_{Cy3}$ or $I_{Cy5}$ approached zero. These adjusted FRET traces and their HMM fittings were manually inspected for correct fittings (Supplementary Fig. 2a–d, Right Panel). Since traces containing Cy3 photobleaching indicated negligible spectral bleed-through to the Cy5 channel, Cy3 bleed-though correction was omitted from $I_{Cy5}$. Intasome aggregates with saturating Cy3 only contributed to a minor Cy5 signal and 0.06 FRET.

### Further selection and categorization of smFRET traces

Intasome injections sometimes created photophysical fluctuations at the beginning of FRET traces. When necessary, these aberrant data were excluded by truncating the traces up to 100 frames. In other cases, the non-specific surface binding of an intasome within a target DNA selection circle resulted in steady Cy3 signals without DNA binding. However, visual inspection of movies allowed us to identify these pseudo-events and elimination by truncation. Only molecules that contained at least 100 frames of data were included in the final analyses.

Traces containing FRET or colocalized Cy3-Cy5 for prolonged time windows (up to minutes) were categorized as STCs. Traces that lack these long events were categorized as TCCs. The data for these two categories were analyzed and presented separately. %STC$_{FRET}$ and their error estimates (σ %STC$_{FRET}$) were calculated from the number of DNAs that showed strand transfer events ($n$) and total number of DNA molecules ($N$), using Eqs. (1) and (2) respectively[66].

$$\% STC_{FRET} = n/N \cdot 100 \tag{1}$$

$$\sigma \% STC_{FRET} = \sqrt{(n/N) \cdot \left(1 - \frac{n}{N}\right)} \cdot 100 \tag{2}$$

### Transition density plots for target capture smFRET traces

A transition density plot was generated by compiling idealized FRET traces resulting from the HMM fittings using an in-house MATLAB program[39]. The peaks in a transition density plot represent transitions from an initial FRET state to a final FRET state and peak height represents the total number of transitions between states[39]. The use of 4 states as an initial guess for HMM, resulted in occasional overfitting of the data and low populated off diagonal peaks.

### Post-synchronized histogram analysis

FRET traces were synchronized to the injection of intasomes to create Post-synchronized histogram plots using an in-house MATLAB program. The assigned 0 FRET corresponding to the intasome unbound state was eliminated for clarity. For 100 ms resolution experiments, 0.01 FRET bins and 300 ms time bins were used to construct Post-synchronized histograms. For 1 s experiments, 0.01 FRET bins and

910 ms time bins were used to construct Post-synchronized histograms.

### smFRET histograms

smFRET histograms were built with 0.2 FRET bins by molecule and time-averaging the >0 FRET states in the traces. The heights of the bins (counts) depend on the number of target DNA molecules included in the analysis. For TCC histograms, this dependency was eliminated by recalculating the counts per DNA. To directly compare between Cy3-PFV experiments the histograms were re-normalized to the number of events in the pseudo-FRET -0.06 peak. This was done by integrating the area under the pseudo-FRET peak to calculate the raw counts and then dividing the whole histogram by that number (Supplementary Fig. 6).

All the smFRET histograms were prepared and fitted using Igor Pro 8 (WaveMetrics). The following Gaussian equation was used for fittings.

$$y = \sum_{i=1}^{n} 1/\sqrt{2\pi\sigma_i^2} e^{\frac{(x-x_{0,i})^2}{2\sigma_i^2}} \tag{3}$$

In most cases a single Gaussian ($n = 1$) fit the data well. Occasionally $n = 2$ or $n = 3$ yielded the best fits.

### Dwell time and transition count histograms

The dwell times of the bound ($t_{on}$), and unbound ($t_{off}$) states were extracted from the TCC traces using the HMM fittings, as shown in Supplementary Fig. 8a. Considering a FRET threshold of 0.1, FRET > 0.1 was defined as bound, and 0 FRET was defined as unbound (Supplementary Fig. 8a). The dwell times were binned according to the web-based bin optimization algorithm (https://www.neuralengine.org//res/histogram.html), and histograms were generated in Igor Pro 8 (WaveMetrics). Single exponential functions were used to calculate the average $\tau_{on}$ and $\tau_{off}$, where $A_1$ and $A_2$ are constants.

$$Counts_{bound} = A_1 e^{-t_{on}/\tau_{on}} \tag{4}$$

$$Counts_{unbound} = A_2 e^{-t_{off}/\tau_{off}} \tag{5}$$

The transition counts for TCC traces were calculated using a FRET threshold of 0.1. Increasing FRET values that crossed the threshold was defined as transitions. Transition counts from individual traces were binned with 1-transition bins to build histograms in Igor Pro 8 (WaveMetrics).

### Overall strand transfer (ST) times

Strand transfer time ($t_{ST}$) is defined as the time from the intasome injection to the first frame that a stable FRET or Cy3 signal appear (Supplementary Table 6). A MATLAB script was used to extract $t_{ST}$ from individual traces. The mean ($\bar{t}_{ST}$) and standard deviation ($\sigma_{ST}$) for a given target DNA was calculated as,

$$\bar{t}_{ST} = \frac{\sum_{i=1}^{N} t_{ST}}{n} \tag{6}$$

$$\sigma_{ST} = \sqrt{\frac{1}{n-1} \sum_{i=1}^{n} (t_{ST} - \bar{t}_{ST})^2} \tag{7}$$

Where $n$ is the number DNA molecules that showed strand transfer events.

## Ensemble integration assay

This assay was performed using the standard protocol described previously[24,27]. Briefly, 25 nM PFV intasome, 50 ng of 3 kb supercoiled (SC) plasmid DNA (pGEM-T Easy, Promega), 10 mM Bis-tris propane, pH 7.5, 110 mM NaCl, 5 mM MgSO$_4$, 4 µM ZnCl$_2$, and 10 mM DTT in 15 µl total volume were incubated at 37 °C for 5 min. The reactions were terminated by adding 0.1% SDS, 2.5 mM EDTA, 1 mg/ml proteinase K and incubated at 55 °C for an hour. The products were mixed with 5% glycerol before resolving on a 1% agarose gel in 1X TAE at 105 V for an hour. Gels were stained with 0.1 µg/mL ethidium bromide and scanned on a Sapphire Biomolecular Imager (Azure Biosystems). Unreacted SC plasmid and linear concerted integration products were quantified with AzureSpot software (Azure Biosystems) as described previously[24,27] and presented in Supplementary Fig. 1.

## Integration site mapping experiments

Integration site mapping was performed as described previously[62] using target DNAs shown in Supplementary Table 1. Briefly, 10 nM unlabeled PFV intasomes, 5 nM target DNA, 30 mM Bis-tris propane, pH 7.5, 110 mM NaCl, 2 mM MgSO$_4$, 4 µM ZnCl$_2$, and 10 mM DTT in 15 µl total volume was incubated at 37 °C for 5 min. Reactions were terminated by adding 0.1% SDS, 2.5 mM EDTA, 1 mg/ml proteinase K and incubating at 55 °C for an hour. Deproteinated samples were denatured by heating to 95 °C with 50% formamide for 10 min, then placed on ice. Sequencing ladders were generated using Thermo Sequenase Dye Primer Manual Cycle Sequencing Kit according to the manufacturer's directions[67] with pDrive-601NPS[40] as the template and Cy5, Alexa488 or Cy3 labeled oligos complementary to the 5′ end of 601NPS as primers. 1 pmol primer and 1 pmol template per reaction were used. Annealing temperature was 55 °C with 45 cycles. Reactions were diluted with an equal volume of formamide, heated to 75 °C for 10 min and stored at −20 °C. Products were resolved on 0.8 mm 8% Acrylamide:Bis 19:1/7 M Urea PAGE gels in 1X TBE at 40 W for various times. Gels were scanned on a Sapphire Biomolecular Imager and quantified using AzureSpot software (Azure Biosystems). Alignment of the products with the sequencing ladder was used to determine the integration sites. The integration efficiency was calculated as the fractional intensity of a band relative to the lane. Reactions without INTs were used as controls.

The site mapping experiments comparing 1nt gap (5′-OH) DNAs with alternative placement of the Cy5 fluorophore were performed as described above with the following modifications in the reaction conditions. 100 nM PFV intasomes assembled with shorter vDNA and Cy3 labeling of the transferred strand [Cy3-PFV(TS), Supplementary Table 1] was reacted with 25 nM DNA targets to increase the yield of integration products. In addition, the reaction buffer was supplemented with 5 nM PCA to improve the lifetime of the intasomes[3]. The PAGE separation of resultant products, the integration site determination and imaging were performed as above.

## $R_{Cy3 \to Cy5}$ and $E_{TCC \text{ or } STC}$ estimations from the crystal structures

The PFV intasome TCC and STC structures (PDB code: 3OS2 and 3OS0, respectively) were downloaded from the PDB data base. To measure the distances ($R_{Cy3 \to Cy5}$) between the fluorophores in vDNAs and the target DNAs, we extended the target DNA chains in both crystal structures due to absence of some base pairs in the structures compared to the target DNA used in the smFRET experiments. Double stranded DNA (5′ CCCGAG 3′) was modeled and virtually ligated to the target DNA of the intasome structures (Discovery Studio, BIOVIA). The novel structure was used to estimate distances ($R_{Cy3 \to Cy5}$) between the labeled bases for various complexes (PyMol 2.1, Schrödinger). These $R_{Cy3 \to Cy5}$ were converted to FRET efficiencies ($E$) using the Förster equation, assuming complete free rotation of

the dyes the orientation factor ($\kappa^2$) is 2/3 and the Förster radius for Cy3→Cy5 ($R_{0,Cy3 \to \backslash Cy5}$) is 6 nm[34].

$$E = \frac{1}{1 + \left(\frac{R_{Cy3-Cy5}}{R_{0,Cy3-Cy5}}\right)^6} \tag{8}$$

The presence of a FRET donor on both vDNAs results in two FRET states. In this case the expected FRET ($E_{TCC}$ or $E_{STC}$) was determined as the arithmetic mean of the two states.

## Reporting summary

Further information on research design is available in the Nature Portfolio Reporting Summary linked to this article.

## Data availability

Primary single molecule imaging data will be made available upon practical request as the files are generally quite large. Two files from the database PDB under accession codes 3OS2 and 3OS0 were used to estimate FRET efficiencies. Source data are provided in the Source Data File. Source data are provided with this paper.

## Code availability

MATLAB code has been deposited at Zenodo under accession code 8411026.

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

## Acknowledgements

The authors would like to thank Ryan Messer and Yow Yong Tan for technical assistance; and Rob Levendosky, and Gregory Bowman (John Hopkins University) for providing protocols for the sequencing ladder. This work was supported by the National Institutes of Health (GM150003 to K.E.Y and R.F.).

## Author contributions

G.S., R.F. and K.E.Y. conceived, designed, and analyzed the smFRET experiments. J.L. wrote and updated the MATLAB code used for the smFRET data analysis. G.S., A.G., R.F. and K.E.Y. conceived, designed and analyzed the integration site mapping studies. G.S., J.L., A.G., R.F. and K.E.Y. wrote and revised the manuscript.

## Competing interests

The authors declare no competing interests.
