## [Peer Review File · Nature Communications]

DNA Strand Breaks and Gaps Target Retroviral Intasome Binding and IntegrationREVIEWER COMMENTS

Reviewer #1 (Remarks to the Author):

Integrase inserts both ends of retroviral DNA into a host chromosome. To do that, it assembles into a multimer on viral DNA ends forming a nucleoprotein complex, which we call the intasome. The intasome then goes on to bind target DNA (chromosomal DNA) for integration. Intasomes from several retroviruses (PFV, RSV, HTLV, HIV, etc) have been studied in vitro, of which the PFV intasome is the best behaved. By now has been characterized using X-ray crystallography, cryo-EM, and other biophysical techniques.

In the current work, Yoder and colleagues examined effects of various defects in target DNA structure on PFV intasome binding and activity. They show the nicks and gaps make the DNA very attractive to the intasome, while some other defects do not. Overall, although the main results are interesting, their relevance to integration during retroviral infection in cells is completely unclear. Do the authors expect that a significant proportion of integration events occur via nicks and gaps? If so, their own data suggest that such events should not be productive, as they would leave to single-end (half-site) integration.

Specific comments:

1) The apparent disassembly of stalled integration events (lines 266-276). This observation is potentially more important than the main story. It makes a lot of sense: the conformational stress of distorted tDNA was proposed to guide the reaction forward during retroviral integration and DNA transposition (see Mu phage transposase structures, for example). Next to a gap, there is no conformational stress, so the reversal is more likely to occur. Since it will re-generate an active intasome, it could rescue the virus. I wonder if that could occur in vivo under suboptimal concentrations of strand transfer inhibitors, or under other conditions when only one viral DNA end can be inserted.

2) Figure 5d (STC) shows a significant fraction of low-FRET events, by contrast Figure 5c (TCC). Are these disassembled STCs?

3) Is the difference in "dynamics" of TCC and STC observed in Figure 5 is intriguing and potentially important. Is it statistically significant, as presented?

4) Line 251: "Because PFV integration events are separated by 4bp..." will not be understood as intended by anyone outside of integrase field.

5) Lines 357-362: Stably bent DNA is not necessarily more flexible. It is clear why intasome would select a gap or a nick (and in some cases DNA end). By contrast, there is no reason to expect that the intasome should preferentially bind to any DNA bend, especially if it does not have a good geometric fit.

Reviewer #3 (Remarks to the Author):

In this manuscript, the authors have created a FRET donor labeled PFV intasome and an array of acceptor-labeled test substrates with varying DNA lesions. Using TIRF-based surface-immobilized smFRET they have assayed the activity of these intasomes on varying target substrates. First by measuring the degree to which the immobilized substrate can transiently capture intasomes (target capture complex), and by investigating FRET activity where the intasome is stably bound (strand transfer complex).

The data show that intasomes have a strong preference for binding substrates with flexibility-conferring lesions such as gaps and nicks, however, lesions that preserve the continuity of the ribose-phosphate backbone (but are otherwise flexible) such as 8-oxo-G, mismatches, or nucleotide insertions do not show activity. This presents an explanation for why retroviruses have limited integration in systems with *ogg1* and *mutyh* mutations. Analysis of prolonged events, which the authors identify as stable strand transfer complexes, demonstrates a similar trend as the more transient binding events.

The STC events were mapped using denaturing PAGE, and the authors thoroughly identify reaction products by nucleotide length. Demonstrating the effects 5'-P on transfer locations and confirming preferences for nicked and gapped substrates.

Finally, the authors used a FRET-labeled intasome with a photobleached target to assess the structural dynamics during the integration of a 1 nt gap. The stable complexes exhibited a much narrower distribution of FRET efficiencies than the transient binding events, which the authors conclude to mean that the stable complex is more rigid than the target capture complex.

Overall the manuscript and the work described are interesting and should be suitable for Nat Commun once the authors address the concerns listed below.

Problems:

The main body of the text makes a lot of combined references to both main figures and extended figures simultaneously and is not always clear which figs are backing up which claims. May help readability if this is broken up a little given just how many figures have been put in the extended.

There are quite a lot of post-synchronized histograms in the main body but the time component of these is rarely commented on, and in the majority of cases the number of events appears to be consistent throughout the time course anyway. Given that the purpose of these PSHs is to construct 1D histograms, it may help if 1D histograms only are shown in the main body and PSHs are kept in the extended figure section.

Line 114- It is claimed that the number of pseudo FRET events per condition does not vary, and this is used as a justification for utilizing it as a baseline for normalization. However, the data in extended data tables 2 and 3 suggest this isn't the case. If the conditions have approximately 500 observations each and the mean value of n/N is 0.6, then the expected standard deviation would be 0.022. However, the conditions have a standard deviation of 0.110 suggesting they differ from one another significantly. This extra variance could potentially be explained by varying concentrations of the Cy3-PFV between conditions, which is understandably difficult to control for at single-molecule concentrations. If this is the case then the pseudo-FRET events are a rational baseline against which to normalize the real events, however, it does call into question the validity of the measurements of $\tau(\text{on})$ which are not normalized to any proxy for concentration.

lines 174/175 refer to a 2 nt gap (3'-OH) substrate. This is probably a typo and meant to be 5'

In fig1a (and 5a) the schematic of the immobilized substrate is shown with an AF488 tag. This tag does not seem to be used for any purpose in the TIRF experiments. The main reason I can think of would be to ensure that the substrates used in the single-molecule experiments remain identical to those used in the PAGE experiments, but the authors should clarify this.

In line 334 the authors refer to the STC FRET distribution as being 'relatively narrow' compared to the TCC distribution ($\sigma = 0.06$ vs 0.10). However, the STC data used in this comparison is at a 1 s frame-rate, which will naturally produce a tighter FRET distribution (due to more photons per frame). In fact, at the 100 msec frame rate done as a control, the distribution widens to $\sigma = 0.09$ which is not considerably tighter than the TCC distribution. When examining extended data figures 12e and 12f,

the STC distribution does appear to have a much-reduced tail compared to the TCC. I suggest that the conclusion the authors are trying to draw from the experiments in Figure 5 would be much better supported by presenting a combination of edf12e and 12f overlayed on each other as part of Figure 5.

Line 339 The authors attempt to compare the FRET efficiencies measured against predictions from the crystal structures. But it is unclear whether the FRET efficiencies in this paper are adequately corrected for conversion to distance. Line 545 says that Cy3 bleed-through correction was omitted due to being negligible, however, donor-only molecules have an apparent $E = 0.06$. Reference 40 is given on line 516 as a description of the software used, this paper described direct excitation correction (unclear if this was done) but does not describe any form of correction for detection efficiencies (i.e., gamma correction) which is known to be a large source of error in determining absolute FRET efficiencies. Furthermore, the FRET efficiency predicted from the crystal structure is calculated from a base-to-base distance rather than modeling the space explored by the diffusing dye, something which is also known to strongly affect FRET efficiencies. Attempting to draw conclusions by comparing an uncorrected FRET efficiency to a simple base-to-base distance with no dye modeling is not wise considering that both are likely to be very inaccurate. Whilst correcting smFRET data for detection efficiencies is difficult without ALEX, the authors could perhaps strengthen this point by predicting the FRET efficiencies from the crystal structures using dye modeling software such as the nano-positioning system from the Michaelis lab or the FRET positioning and screening tool from the Seidel lab.

(Non-truncated Reviewer Comments are shown in red)

Reviewer #1

General:

“Integrase inserts both ends of retroviral DNA into a host chromosome. To do that, it assembles into a multimer on viral DNA ends forming a nucleoprotein complex, which we call the intasome. The intasome then goes on to bind target DNA (chromosomal DNA) for integration. Intasomes from several retroviruses (PFV, RSV, HTLV, HIV, etc) have been studied in vitro, of which the PFV intasome is the best behaved. By now has been characterized using X-ray crystallography, cryo-EM, and other biophysical techniques.

Overall, although the main results are interesting, their relevance to integration during retroviral infection in cells is completely unclear. Do the although the main results are interesting, their relevance to integration during retroviral infection in cells is completely unclear. Do the authors expect that a significant proportion of integration events occur via nicks and gaps? If so, their own data suggest that such events should not be productive, as they would leave to single-end (half-site) integration.”

We have significantly revised the discussion to clarify the connection(s) between the observations present in our manuscript with productive retroviral integration in cells. Reviewer #1 is correct that the principal half-site integration events are most likely to be non-productive as they will be repaired/eliminated by double strand break (DSB) repair. However, the increased TCC events that ultimately lead to the frequent half-site integration events, also lead to an increased number of rarer (as much as 5%) concerted integration events. As outlined in the revised discussion, these events could account for the genetic effects of BER pathway mutations on the frequency of retroviral integration as well as the changes in consensus integration site sequence preferences.

Importantly, regardless of the effect(s) on productive infection it is essential to recognize that our studies present the first observation of a targeted intasome integration and provide a foundation for the design of related configurations that might improve retroviral targeting.

Specific comments:

- 1) *“The apparent disassembly of stalled integration events (lines 266-276). This observation is potentially more important than the main story. It makes a lot of sense: the conformational stress of distorted tDNA was proposed to guide the reaction forward during retroviral integration and DNA transposition (see Mu phage transposase structures, for example). Next to a gap, there is no conformational stress, so the reversal is more likely to occur. Since it will re-generate an active intasome, it could rescue the virus. I wonder if that could occur in vivo under suboptimal concentrations of strand transfer inhibitors, or under other conditions when only one viral DNA end can be inserted.”*

We have never observed reversal of the strand transfer reaction, which would be quite obvious in our single molecule imaging systems. It is possible that such events might occur over much longer time periods. However, photobleaching and/or fluorophore lifetime may become limiting factors for such long-term observations. We appreciate the experimental suggestions by Reviewer #1 but consider these beyond the scope of the present manuscript.

- 2) *“Figure 5d (STC) shows a significant fraction of low-FRET events, by contrast Figure 5c (TCC). Are these disassembled STCs?”*

The very low FRET events shown in Fig. 5d (STC) are the result of STC event by intasomes that contain two Cy3-vDNAs. These occur with predictable statistics based on the method of Cy3/Cy5-PFV intasome assembly where we start with a 50:50 mix of Cy3-vDNA and Cy5-vDNA substrates. STCs with two Cy3-vDNAs produce increased Cy3 emission intensity that bleeds into the Cy5 channel resulting in a pseudo-FRET (low-FRET). We have revised the text to better explain this observation.

- 3) *“Is the difference in “dynamics” of TCC and STC observed in Figure 5 is intriguing and potentially important. Is it statistically significant, as presented?”*

We have revised Fig. 5c and 5d such that they reflect an identical acquisition frame rate (1 sec). Moreover, we have revised Extended Fig. 12e to show the faster acquisition frame rate for TCC events (100 msec) and formulated a new panel (Extended Fig. 12f) showing an overlay histogram comparison of TCC and STC FRET events collected at 100 msec frame rate.

We found that the mean and width of a Gaussian fit of the TCC and STC histograms was not significantly different. However, the overlay of these plots still shows an extended low FRET tail in the TCC histogram (Extended Data 12f). This extended low FRET tail is consistent with the hypothesis that TCC events display modestly increased dynamics compared to STC events. We note an even broader and extended tail with TCC events collected at 1 sec frame rate. But the 1 sec frame rate exceeds the average τ_{on} , while 100 msec is ~4-fold below the average τ_{on} for a 1nt GAP(5'-OH) target DNA substrate (Fig. 2a). Thus, the broadened FRET and extend tail at 1 sec frame rate in Fig. 5c almost certainly results from significant numbers of time-averaged FRET events. It is also possible that some fraction of the 100 msec frame rate events may reflect time-averaged FRET events. With these caveats in mind, we have substantially moderated our conclusion that TCC events display increased intasome conformational dynamics compared to STC events.

- 4) *“Line 251: “Because PFV integration events are separated by 4bp...” will not be understood as intended by anyone outside of integrase field.”*

We have revised the text to better describe the nucleotide separation between catalytic PFV strand transfer events.

- 5) *“Lines 357-362: Stably bent DNA is not necessarily more flexible. It is clear why intasome would select a gap or a nick (and in some cases DNA end). By contrast, there is no reason to expect that the intasome should preferentially bind to any DNA bend, especially if it does not have a good geometric fit.”*

Reviewer #1 is correct that a stable bend does not imply flexibility, or the ability to provide a good geometric fit for intasome binding. That said, previous work from our group has demonstrated that a G/T mismatch is intrinsically more flexible than duplex DNA, and the differences between dynamic mispair-dependent backbone flexibility is likely the biophysical basis for the range and relative efficiency of recognition by mismatch binding proteins (Mazurek et al., *PNAS* 106:4177, 2009). Moreover, structural studies have also shown that a +T mismatch induces a fairly stable 22° bend in the DNA (Joshua-Tor, et al., *J. Mol. Biol.* 225:397, 1992). For mismatch binding proteins, the dynamic backbone flexibility and/or stable bend is exploited to capture and/or induce a transiently stable 40-60° bend at the mispair during the binding progression. We think Reviewer #1 will agree that it was worth testing whether PFV intasomes could also capture this intrinsic flexibility/bend. With the answer being no, the obvious question is why – leading to the hypothesis that it is not the bent DNA *per se* but the correspondingly altered DNA twist that provided a better

geometric fit and enhances binding. We have modified the verbiage to reflect the nuances of this idea.

Reviewer #3

General:

"In this manuscript, the authors have created a FRET donor labeled PFV intasome and an array of acceptor-labeled test substrates with varying DNA lesions. Using TIRF-based surface-immobilized smFRET they have assayed the activity of these intasomes on varying target substrates. First by measuring the degree to which the immobilized substrate can transiently capture intasomes (target capture complex), and by investigating FRET activity where the intasome is stably bound (strand transfer complex).

*The data show that intasomes have a strong preference for binding substrates with flexibility-conferring lesions such as gaps and nicks, however, lesions that preserve the continuity of the ribose-phosphate backbone (but are otherwise flexible) such as 8-oxo-G, mismatches, or nucleotide insertions do not show activity. This presents an explanation for why retroviruses have limited integration in systems with *ogg1* and *mutyh* mutations. Analysis of prolonged events, which the authors identify as stable strand transfer complexes, demonstrates a similar trend as the more transient binding events.*

The STC events were mapped using denaturing PAGE, and the authors thoroughly identify reaction products by nucleotide length. Demonstrating the effects 5'-P on transfer locations and confirming preferences for nicked and gapped substrates.

Finally, the authors used a FRET-labeled intasome with a photobleached target to assess the structural dynamics during the integration of a 1 nt gap. The stable complexes exhibited a much narrower distribution of FRET efficiencies than the transient binding events, which the authors conclude to mean that the stable complex is more rigid than the target capture complex.

Overall, the manuscript and the work described are interesting and should be suitable for Nat Commun once the authors address the concerns listed below."

Specific Comments:

- 1) *"The main body of the text makes a lot of combined references to both main figures and extended figures simultaneously and is not always clear which figs are backing up which claims. May help readability if this is broken up a little given just how many figures have been put in the extended."*

We have included additional information when referencing Extended Data to indicate which material supports the preceding sentence, paragraph, or conclusion. We believe these additions will enhance readability and further focus the associated data.

- 2) *"There are quite a lot of post-synchronized histograms in the main body but the time component of these is rarely commented on, and in the majority of cases the number of events appears to be consistent throughout the time course anyway. Given that the purpose of these PSHs is to construct 1D histograms, it may help if 1D histograms only are shown in the main body and PSHs are kept in the extended figure section."*

We have elected to retain the time component of the post-synchronized histograms since they clearly illustrate the differences in kinetic accumulation of TCC and STC events. These

observations demonstrate that TCC events begin immediately upon injection of PFV intasomes, but only a fraction of these binding events is converted to STC events that appear at later time points. We note that showing these kinetic examples is also helpful for the reader in discriminating *bona fide* binding events from spurious interactions and/or aggregates.

- 3) *“Line 114- It is claimed that the number of pseudo FRET events per condition does not vary, and this is used as a justification for utilizing it as a baseline for normalization. However, the data in extended data tables 2 and 3 suggest this isn't the case. If the conditions have approximately 500 observations each and the mean value of n/N is 0.6, then the expected standard deviation would be 0.022. However, the conditions have a standard deviation of 0.110 suggesting they differ from one another significantly. This extra variance could potentially be explained by varying concentrations of the Cy3-PFV between conditions, which is understandably difficult to control for at single-molecule concentrations. If this is the case then the pseudo-FRET events are a rational baseline against which to normalize the real events, however, it does call into question the validity of the measurements of $\tau(\text{on})$ which are not normalized to any proxy for concentration.”*

Reviewer #3 is correct in his/her calculations of the standard of deviations. Importantly, the aggregates that contribute to pseudo-FRET events arise spontaneously in spite of our best biochemical conditions, exhibit different sizes and produce different emission intensities, and because they are formed from intasomes may slightly alter the concentration of free Cy3-PFV intasomes that contribute to TCC events. Thus, Reviewer #3 is also correct in his/her assumptions that the free concentrations of Cy3-PFV intasomes may vary between experiments, further supporting our use of the *“pseudo-FRET events (as) a rational baseline against which to normalize the real events”* as he/she suggested. These comments by Reviewer #3 also included a typo since the modest differences in Cy3-PFV intasome concentrations will affect the τ_{off} (which is equal to $1/k_{\text{on}}$) not the τ_{on} (which is equal to $1/k_{\text{off}}$). We have now revised this section to point out the numerous experimental issues with τ_{off} and indicate that the τ_{on} “generally reflect(s) the normalized target DNA binding efficacy” rather than providing a quantitative accounting.

- 4) *“lines 174/175 refer to a 2 nt gap (3'-OH) substrate. This is probably a typo and meant to be 5”*

Typo corrected.

- 5) *“In fig1a (and 5a) the schematic of the immobilized substrate is shown with an AF488 tag. This tag does not seem to be used for any purpose in the TIRF experiments. The main reason I can think of would be to ensure that the substrates used in the single-molecule experiments remain identical to those used in the PAGE experiments, but the authors should clarify this.”*

An additional sentence has been added to clarify the consistent inclusion of the Alexa488 fluorophore with previous target DNA substrates as well as its potential value in DNA localization.

- 6) *“In line 334 the authors refer to the STC FRET distribution as being 'relatively narrow' compared to the TCC distribution ($\sigma = 0.06$ vs 0.10). However, the STC data used in this comparison is at a 1 s frame-rate, which will naturally produce a tighter FRET distribution (due to more photons per frame). In fact, at the 100 msec frame rate done as a control, the distribution widens to $\sigma = 0.09$ which is not considerably tighter than the TCC distribution. When examining extended data figures 12e and 12f, the STC distribution does appear to have a much-reduced tail compared to the TCC. I suggest that the conclusion the authors are trying to draw from the experiments in Figure 5 would be much better supported by presenting a combination of edf12e and 12f overlaid on each other as part of Figure 5.”*

See Reviewer #1 *Specific Comment 3*) above. In short, we have modified Fig. 5c,d to include only 1 sec frame rate data as an “equivalent” comparison. The 100 msec frame rate data is now included in Extended Fig. 12e, and we have overlaid the 100 msec TCC and STC data for a better comparison as suggested by Reviewer #3 (Extended Data Fig. 12f). We found that the difference in the mean and width of a Gaussian fit of these two curves was not statistically different. However, we still noted an evident tail of lower FRET TCC events compared to STC events at the 100 msec frame rate, consistent with the hypothesis that TCC events are likely to be more dynamic. However, with these additional caveats in mind, we have substantially moderated the conclusions that the TCC may be more dynamic than the STC.

7) *“Line 339 The authors attempt to compare the FRET efficiencies measured against predictions from the crystal structures. But it is unclear whether the FRET efficiencies in this paper are adequately corrected for conversion to distance. Line 545 says that Cy3 bleed-through correction was omitted due to being negligible, however, donor-only molecules have an apparent $E = 0.06$. Reference 40 is given on line 516 as a description of the software used, this paper described direct excitation correction (unclear if this was done) but does not describe any form of correction for detection efficiencies (i.e., gamma correction) which is known to be a large source of error in determining absolute FRET efficiencies. Furthermore, the FRET efficiency predicted from the crystal structure is calculated from a base-to-base distance rather than modeling the space explored by the diffusing dye, something which is also known to strongly affect FRET efficiencies. Attempting to draw conclusions by comparing an uncorrected FRET efficiency to a simple base-to-base distance with no dye modeling is not wise considering that both are likely to be very inaccurate. Whilst correcting smFRET data for detection efficiencies is difficult without ALEX, the authors could perhaps strengthen this point by predicting the FRET efficiencies from the crystal structures using dye modeling software such as the nano-positioning system from the Michaelis lab or the FRET positioning and screening tool from the Seidel lab.”*

We have eliminated any discussions of predicted versus measured FRET, since these comparisons are complex, the models are imperfect, and most require factors that are difficult to quantitatively acquire from our systems. Moreover, such comparisons are not essential for the data and conclusions presented in the manuscript.

REVIEWERS' COMMENTS

Reviewer #1 (Remarks to the Author):

The manuscript has improved in revision, although I feel the relevance of the in vitro PFV study to what is happening during HIV integration in cells remains tenuous. That said, the work elucidates interesting and novel aspects of the interaction between the retroviral integration machinery and target DNA, specific defects in which can act as intasome sinks.

Reviewer #2 (Remarks to the Author):

The authors have addressed my concerns.